# Is *O(log N)* practical?
# Near-Equivalence Between Delay Robustness and Bounded Regret in Bandits and RL

**Enoch H. Kang**[*]
Foster School of Business
University of Washington
Seattle, WA 98195

**P. R. Kumar**
Electrical & Computer Engineering
Texas A&M University
College Station, TX 77843

## Abstract

Interactive decision making, encompassing bandits, contextual bandits, and reinforcement learning, has recently been of interest to theoretical studies of experimentation design and recommender system algorithm research. One recent finding in this area is that the well-known Graves-Lai constant being zero is a necessary and sufficient condition for achieving bounded (or constant) regret in interactive decision-making. As this condition may be a strong requirement for many applications, the practical usefulness of pursuing bounded regret has been questioned. In this paper, we show that the condition of the Graves-Lai constant being zero is also necessary for a consistent algorithm to achieve delay model robustness when reward delays are unknown (i.e., when feedback is anonymous). Here, model robustness is measured in terms of $\epsilon$-robustness, one of the most widely used and one of the least adversarial robustness concepts in the robust statistics literature. In particular, we show that $\epsilon$-robustness cannot be achieved for a consistent (i.e., uniformly sub-polynomial regret) algorithm, however small the nonzero $\epsilon$ value is, when the Grave-Lai constant is not zero. While this is a strongly negative result, we also provide a positive result for linear rewards models (contextual linear bandits, reinforcement learning with linear MDP) that the Grave-Lai constant being zero is also sufficient for achieving bounded regret without any knowledge of delay models, i.e., the best of both the efficiency world and the delay robustness world.

## 1 Introduction

We consider the cost of addressing stochastic and anonymous delayed rewards in *Decision-Making with Structured Observations (DMSO)* [1, 2], which generalizes interactive decision-making problems, such as structured bandits, contextual bandits, and reinforcement learning[2]. In many real-life applications of interactive decision-making problems, stochastic and unknown delays in reward make it challenging to attribute the sequence of observed outcomes to previous decisions. In medical treatments, for example, a doctor cannot easily be sure whether a medical outcome is due to the effect of current treatment or due to some other previously taken treatment's delayed effect. This type of reward delay in decisions is called an 'unknown reward delay' [3] or 'delayed anonymous feedback' [4, 5][3]. Under this setting, the decision maker never observes the period information for which each

---

[*]ehwkang@uw.edu

[2]One can refer to Appendix A (or [2, 1] for a more comprehensive, detailed description) to see how bandit, contextual bandit, and episodic reinforcement learning problems can be described as DMSO problems.

[3]Most previous studies focus on delayed, anonymous, and *aggregated* (DAAF) feedback, where only the observe sum of the rewards arriving at each episode is observed. Here, we consider impossibility results for

38th Conference on Neural Information Processing Systems (NeurIPS 2024).

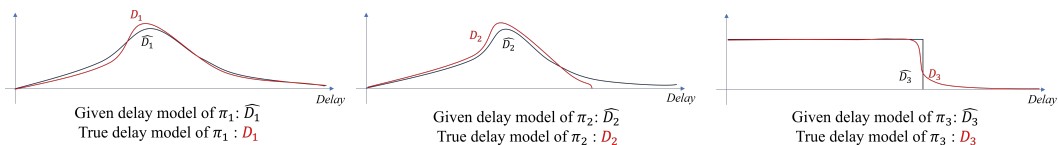

Figure 1: Examples of misspecification of the reward delay model of decisions $(\pi_1, \pi_2, \pi_3)$

reward corresponds to, even after it receives the delayed reward at the later time step. As it is not obvious which decision caused each observed reward, reward attribution becomes a challenge.

Some knowledge (e.g., mean) of the probabilistic distribution of each decision's reward delay, combined with the careful design of algorithms, may help to resolve this reward attribution problem under stochastic and anonymous delayed rewards [4]. However, those delay models themselves may be misspecified [6]. Therefore, whether we can design an algorithm that is robust to model misspecification becomes a main concern in the problems with stochastic and anonymous delayed rewards.

One of the most widely used concepts of model misspecification in the robust statistics literature is $\epsilon$-*robustness* [7]. Given a parameter $\epsilon > 0$ and true distribution $D$, a model distribution $\widehat{D}$ is called an $\epsilon$-(general) contamination of $D$ if $d_{TV}(D, \widehat{D}) \leq \epsilon$, where $d_{TV}$ denotes the total variation distance function[4]. Figure 1 illustrates some examples of $\epsilon$-contamination of the delay models. As $\epsilon$-robustness is also one of the weakest (i.e., least adversarial) and most elementary notion of robustness [8], the first question on an algorithm's delay robustness will be, "up to which $\epsilon$ the algorithm's properties are robust to $\epsilon$-contamination of delay model misspecification?".

In this paper, we prove that no consistent (i.e., uniformly sub-polynomial regret) algorithm for DMSO can be designed to be robust to $\epsilon$-contamination of delay model misspecification unless DMSO's Graves-Lai constant [9, 2, 1] being zero. While this is a strong negative result, we also provide a positive result for linear DMSO problems (linear contextual bandit, reinforcement learning with linear MDPs), showing that the Graves-Lai constant being zero [9, 10, 11] is sufficient for achieving bounded regret without any knowledge of delay models. As the Graves-Lai constant being zero holds if and only if we can achieve bounded regret [2, 1, 10, 11], the results in this paper strongly motivate the practical usefulness of designing learning systems where we can achieve bounded regret.

## 1.1 Related work

While no previous work has studied the link between consistent algorithms and *robustness to delay distribution misspecification* when reward delays are unknown (=anonymous) in problems related to DMSO (e.g, bandit problems and reinforcement learning problems), there has been other work looking at different flavors of delayed anonymous rewards. There were prior studies on delayed rewards [12, 13, 14, 15], but [4] was the first to formalize the unknown stochastic reward delays assumption in interactive decision making problems, which led to the literature on stochastic delayed, anonymous and aggregated feedback (DAAF). While [4] provides a consistent algorithm for stochastic bandits that does not require any knowledge of delay distributions, it requires a strong assumption that the mean of delay distribution is precisely known, which cannot be achieved under $\epsilon$-contamination however small $\epsilon$ is[5] (In Section 4, we show that no algorithm can be consistent when $\epsilon > 0$). [16] provides another consistent algorithm for stochastic bandits that also does not require any knowledge of delay distributions and improves [4], but it requires a different assumption that the delayed reward feedback exactly associates the reward and the arm and therefore rewards are not anonymous; [17, 18] make similar assumption for episodic reinforcement learning problems with stochastic delays.

---

the delayed, anonymous, and *non-aggregated* feedback, where we observe each delayed anonymous reward separately.

[4]The total variation distance $d_{\mathrm{TV}}(\nu, \upsilon)$ is defined as $\frac{1}{2}\|\nu - \upsilon\|_1 = \sup_{E \in \Sigma} |\nu(E) - \upsilon(E)|$, where $\Sigma$ stands for the measurable sets on which two distributions $\nu$ and $\upsilon$ are defined.

[5]For example, for the family of distributions with $k$-th moment bounded by 1 for $k \geq 2$, $\epsilon$-contamination in delay distribution, i.e., $d_{TV}\left(D, \hat{D}\right) > \epsilon$, implies $\left|\mathbb{E}\left[D\right] - \mathbb{E}\left[\hat{D}\right]\right| > k\epsilon^{1-1/k}$ (See Assumption 4.2 for more).

While our work centers on delayed anonymous rewards—where the learner cannot associate delayed rewards with the actions that generated them—there exists a parallel line of research addressing non-anonymous delays, in which such associations are possible. In this context, several studies have proposed algorithms that account for unrestricted or unbounded delays. [19] developed algorithms for nonstochastic multiarmed bandits with unrestricted delays, achieving robust regret bounds by employing a skipping strategy to manage excessively delayed feedback. [20] adapted Thompson sampling to handle multiarmed bandits with unrestricted delays, extending its applicability to delayed feedback settings without assuming bounded delays. [21] derived near-optimal regret bounds for adversarial MDPs with delayed bandit feedback, addressing the challenges posed by feedback delays in adversarial environments. [22] proposed an optimal algorithm for adversarial bandits experiencing arbitrary delays, establishing regret bounds that hold even when delays are extensive. Building on this foundation, [23] introduced a "best-of-both-worlds" algorithm that improves upon [22] by providing both adversarial guarantees and near-optimal stochastic performance without requiring prior knowledge of the maximal delay.

## 2 Preliminaries

### 2.1 Decision-Making with Structured Observations

The DMSO problem framework generalizes many problems such as bandit problems, contextual bandit problems, and episodic reinforcement learning problems. DMSO is characterized by the environment and its learning protocol, which is described as follows:

○ The environment of a DMSO problem framework is specified as a tuple $(\Pi, \mathcal{R}, \mathcal{O}, \mathcal{F})$, where $\Pi$ denotes the decision space, $\mathcal{R}$ denotes the reward space, $\mathcal{O}$ denotes the observation space, and $\mathcal{F} = \prod_{\pi \in \Pi} \mathcal{F}_\pi$ denotes the model class where $\mathcal{F}_\pi \subseteq \triangle_{\mathcal{R} \times \mathcal{O}}$ (Here, $\triangle_E$ notation means the space of all possible probability distributions over a set $E$).[6] We use $f_\pi$ to refer to an element of $\mathcal{F}_\pi$, with $f_\pi$ being the $\pi$-coordinate of $f \in \mathcal{F}$. A ground-truth model $f^\star \in \mathcal{F}$ governs the rewards and the observations based on the decisions made in the rounds. While $f^\star$ is unknown to the learner, it is typically assumed that a set $\mathcal{F}$ that includes $f^\star$ is known to the learner. Formally, we make the following assumption, which is often called the realizability assumption [24, 25, 26].

**Assumption 2.1** (Realizability). The learner has access to the model class $\mathcal{F}$ containing the ground-truth model $f^\star$.

○ The learning protocol for the DMSO problem consists of $n$ rounds. In round $k \leq n$,

1. The learner makes a decision $\pi_k \in \Pi$.
2. A reward $r_k \in \mathcal{R}$ and an observation $o_k \in \mathcal{O}$ are generated, where $(r_k, o_k) \sim f^\star_{\pi_k} \in \mathcal{F}_{\pi_k}$
3. Learner observes $o_k$. If there are reward delays, the learner observes $R_k$, the set of rewards that arrive at the round $k$. $R_k$ is equivalent to $r_k$ only if there are no reward delays.

### 2.2 Learning Algorithm for DMSO

Given that we characterized the DMSO problem framework, we can now describe a learning algorithm for it. Let $h_k$ be the history up to round $k$, i.e., $h_k = \{(\pi_j, R_j, o_j)\}_{j=1}^{k-1}$ where $\pi_j \in \Pi, R_j \in \mathcal{R}, o_j \in \mathcal{O}$ and $\mathcal{H}$ be the set of all possible histories of rounds for $k \geq 1$. A learning algorithm $A$ is defined as an element of $\mathcal{A} \subseteq (\mathcal{H} \mapsto \triangle_\Pi)$, which is a subset of the set of all possible mappings from the history space $\mathcal{H}$ to the set of all possible distributions over $\Pi$. That is, at each round $k$, given the history $h_k \in \mathcal{H}$, a learning algorithm $A \in \mathcal{A}$ chooses $p_k = A(h_k) \in \triangle_\Pi$. The decision at round $k$, $\pi_k$, is sampled from $p_k$. Note that $f \in \mathcal{F}, A \in \mathcal{A}$ and the round $n$ completely determine the stochastic behavior of the learning protocol up to round $n$, i.e., they induce a probability distribution we call $P_{f,n,A}[\cdot]$ over the set of all histories up to round $n$. We also denote the respective expectation by $\mathbb{E}_{f,n,A}[\cdot]$. When the meaning is clear from the context, we use $P_{f,n}[\cdot]$ and $\mathbb{E}_{f,n}[\cdot]$ instead of $P_{f,n,A}[\cdot]$ and $\mathbb{E}_{f,n,A}[\cdot]$.

Given $(r, o) \sim f_\pi$, we denote $\mu_{f_\pi} := \mathbb{E}_{f_\pi}[r]$. Let $\pi_f \in \arg\max_{\pi \in \Pi} \mu_{f_\pi}$ denote an optimal decision for the model $f$. The sub-optimality gap of decision $\pi$ for model $f$ is defined as $\Delta_f(\pi) := \mu_{f_{\pi_f}} - \mu_{f_\pi}$. When the ground truth model is $f$, choosing $\pi_f$ at each round until round $n$ yields the largest value of

---

[6]This model class definition is equivalent to the definition used by Wagenmaker and Foster [1] because if $A$ is a set, countable or not, the Cartesian product $X^A$ is defined to be the collection of all functions $f : A \to X$.

total reward until round $n$. Therefore, we can measure the optimality of an algorithm $A$ until round $n$ in terms of regret, which is defined by

$$\text{Reg}_{f,A}(n) := \mathbb{E}_{A,f,n}\left[\sum_{k=1}^{n}\Delta_f(\pi_k)\right]. \tag{1}$$

## 2.3 Consistent Learning Algorithm's Instance-Dependent Regret Lower Bound

$\text{Reg}_{f,A}(n)$ is a quantity that is dependent on the true model instance $f$. Since the true model $f$ is unknown to the learner a priori, a good learning algorithm must be able to perform well for all possible $f \in \mathcal{F}$. Therefore, one might want to define the goodness of an algorithm by its capability to achieve the minimal value of $\text{Reg}_{f,A}(n)$ among all possible algorithms for all the instances $f \in \mathcal{F}$. However, this is not achievable; a bespoke algorithm that always chooses $\pi_f$ will outperform all possible algorithms for the instance $f$, while suffering linear regret for the instances in $\mathcal{F} \setminus f$.

Since many problems have algorithms that incur sub-polynomial regret for all instances, it is common to exclude algorithms that suffer polynomially increasing regret in some instances. This idea is formalized in the following definition that restricts the space of 'interesting' algorithms.

**Definition 2.2** (Graves and Lai (1997) [9]). A learning algorithm $A$ is called consistent if $\text{Reg}_{f,A}(n) = o\left(n^p\right)$ holds for every $p > 0$ and $f \in \mathcal{F}$.

For DMSO problems, it has been recently shown that any consistent algorithm's instance-dependent regret must satisfy the asymptotic lower bound described in the following theorem [2].

**Theorem 2.3** (Dong and Ma (2022) [2]). *Suppose that there are no reward delays. Then for every instance $f \in \mathcal{F}$, the expected regret of any consistent algorithm $A$ satisfies*

$$\limsup_{n \to \infty} \frac{\text{Reg}_{f,A}(n)}{\ln n} \geq \mathcal{C}(f) = \lim_{n \to \infty} \mathcal{C}(f,n), \tag{2}$$

*where $\mathcal{C}(f,n)$ is the solution to the optimization equation*

$$\mathcal{C}(f,n) \triangleq \min_{w \in \mathbb{R}_+^{|\Pi|}} \sum_{\pi \in \Pi} w_\pi \Delta_f(\pi)$$
$$s.t. \sum_{\pi \in \Pi} w_\pi D_{\text{KL}}(f_\pi \| g_\pi) \geq 1 \ , \ \forall g \in \mathcal{F}(f)^c \tag{3}$$
$$\|w\|_\infty \leq n,$$

*where $D_{KL}$ is the KL divergence and $\mathcal{F}(f) := \{g \in \mathcal{F} \mid \pi_g = \pi_f\}$.*

**Corollary 2.4.** *Let $f^\star \in \mathcal{F}$ be the ground-truth model. For a consistent algorithm to achieve sub-logarithmic regret, $\mathcal{C}(f^\star) = 0$ must hold. That is, achievement of sub-logarithmic regret for all possible instances of $\mathcal{F}$ can be assured a priori only if $\mathcal{C}(f) = 0$ for $f \in \mathcal{F}$.*

**Theorem 2.5** (Wagenmaker and Foster [1]). *Expected regret of order $\mathcal{C}(f^\star) \ln n$ can be achieved for DMSO problems without delays. That is, bounded regret can be assured if $\mathcal{C}(f) = 0$ for $f \in \mathcal{F}$.*

## 2.4 $\epsilon$-contamination and Total Variation Distance

In robust statistics, one of the oldest and the most commonly used concepts for modeling contaminated data is the concept of $\epsilon$-contamination [7]. Given a parameter $0 < \epsilon < 1$ and original distribution $D$, a distribution $X$ is called an $\epsilon$-additive contamination (or Huber contamination) of $D$ if $X$ is a mixture distribution of $D$ and an unknown arbitrary distribution $E$, with their selection probabilities being $(1 - \epsilon)$ for $D$ and $\epsilon$ for $E$. Furthermore, A distribution $X$ is called an $\epsilon$-subtractive contamination of $D$ if it is equivalent to an arbitrary $\epsilon$-probability removal from $D$ (and normalization). Finally, we say that a distribution $X$ is a (general) $\epsilon$-contamination of $D$ if $X$ can be constructed by removing $\epsilon$-probability from $D$ and replacing that $\epsilon$ with equal mass from some arbitrary distribution $E$.

As discussed earlier, the concept of $\epsilon$-contamination is closely related to the concept of total variation distance. Given a space of distributions, the total variation distance, denoted as $d_{TV}(\nu, \upsilon)$, is defined as $d_{\text{TV}}(\nu, \upsilon) := \frac{1}{2}\|\nu - \upsilon\|_1 = \sup_{E \in \Sigma} |\nu(E) - \upsilon(E)|$, where $\Sigma$ stands for the measurable sets on which $\nu$ and $\upsilon$ are defined. It is well known that $d_{TV}$ is a metric that satisfies interesting properties

such as 1) $d_{TV} = 0$ if and only if $\nu = \upsilon$ and 2) $d_{TV} = 1$ if and only if $\nu$ and $\upsilon$ are singular, i.e. there exists $E$ such that $\nu(E) = 1$ and $\upsilon(E) = 0$. It is also well known that the concept of $\epsilon$-contamination is equivalent to the concept of total variation distance [8]; we separately state this property as the following lemma.

**Lemma 2.6** ([8]). *Given a parameter $0 < \epsilon < 1$, a distribution $X$ is an $\epsilon$-contamination of $D$ (and vice versa) if and only if $d_{TV}(X, D) = \epsilon$.*

## 3 Main Model: $\epsilon$-delay Robustness

Denote the true reward delay distributions of each decision $\pi \in \Pi$ by $D_\pi$. Every time $\pi_k$ is determined at each round $k$, $d_k \sim D_{\pi_k}$ is generated along with the generation of $(r_k, o_k) \sim f_\pi$. While $o_k$ is observed immediately at round $k$, $r_k$ is scheduled to arrive at round $d_k + k$. The order of reward arrivals does not necessarily match the order of the reward generation.

As in [3], we assume the *unknown delays* setting throughout the paper. In the unknown delays setting, the delay $d_k$ is not observed, and therefore attributing rewards to the previous decisions becomes a nontrivial problem. One can only guess from which decision the reward just arrived came based on the history of previous decisions and some information about reward delay distributions.

As reward delays are not observed, information we know about reward delay distributions is likely to be misspecified. We model this misspecification as $\epsilon$-contamination of the true delay distribution models $\{D_\pi\}_{\pi \in \Pi}$ resulting in information about $\{\hat{D}_\pi\}_{\pi \in \Pi}$ instead, where $\hat{D}_\pi$ is an outcome of $\epsilon$-contamination of the delay distribution model $D_\pi$, i.e., $d_{TV}(D_\pi, \hat{D}_\pi) \leq \epsilon$. Note that $\epsilon$-contamination of delay distribution models encompasses many possible misspecification of information about delay distribution. For example, it implies misspecification of mean of delay distribution as $d_{TV}(D_\pi, \hat{D}_\pi) > \epsilon$ implies $|\mathbb{E}[D] - \mathbb{E}[\hat{D}]| > 0$ (See discussions in Assumption 4.2 for details).

We now propose the formal definition of robustness in terms of delay distribution knowledge.

**Definition 3.1.** We say that a consistent algorithm is $\epsilon$-delay robust if it is consistent when the given delay distributions are $\epsilon$-contaminations of the true delay distributions.

## 4 Main Results

### 4.1 Negative Result: Delay Robustness Requires $\mathcal{C}(f) = 0$ for all $f \in \mathcal{F}$

Given the definition of $\epsilon$-delay robustness provided in Section 3, the main question is when a consistent, $\epsilon$-delay robust algorithm exists. The answer is quite negative: unless $\mathcal{C}(f^\star) = 0$ holds, no consistent algorithm can be $\epsilon$-delay robust, however small $\epsilon > 0$ is.

**Theorem 4.1.** *Under minor technical assumptions (see Assumptions 4.2-4.5 below), regardless of how small $\epsilon > 0$ is, a consistent learning algorithm can be $\epsilon$-delay robust only if $\mathcal{C}(f^\star) = 0$.*

Theorem 4.1 implies that the concept of consistent algorithm fails even with a very small misspecification of the delay model unless the Graves-Lai constant $\mathcal{C}(f^\star)$ satisfies $\mathcal{C}(f^\star) = 0$. Since we want to design a learning system with existence of a consistent algorithm that works for all instances of $f \in \mathcal{F}$, we need $\mathcal{C}(f) = 0$ for $f \in \mathcal{F}$, which was the necessary and sufficient condition for achieving bounded regret when there were no reward delays (Corollary 2.4 and Theorem 2.5)

The intuition behind the proof of Theorem 4.1 (See Appendix B for details), is as follows. When the reward delay model is precisely known, i.e., when the reward delay model is not contaminated, we might be able to address this challenge by designing a good algorithm that makes the probability of confusion in reward attribution as small as we want. However, in the case of $\epsilon$-contamination of delay models, under minor technical assumptions below (Assumptions 4.2-4.5), we can always provide a delay model contamination that makes the precision of any consistent algorithm's reward attribution no better than $1 - \delta$ for some $\delta > 0$. This leads to reward distribution suffering $\delta$-contamination, which makes impossible to design a consistent algorithm.

The assumptions required for Theorem 4.1 are as follows.

**Assumption 4.2.** For the family of distributions $\mathcal{D}_{r|o} := \{f_\pi(\cdot \mid o) \mid f \in \mathcal{F}, \pi \in \Pi, o \in \mathcal{O}\}$, there exists a function $q(\delta)$ s.t. for $D_1, D_2 \in \mathcal{D}_{r|o}$, $|\mathbb{E}[D_1] - \mathbb{E}[D_2]| \leq q(\delta)$ implies $d_{TV}(D_1, D_2) \leq \delta$.

For some special families of reward distributions, such $q$ that satisfies Assumption 4.2 is known [8]. (Let $k$ be a constant in what follows)

- For the family of Gaussian distributions with standard deviation 1, $q(\delta) = k\delta$.
- For the family of log-concave distributions with standard deviation 1, $q(\delta) = k\delta \log(1/\delta)$.
- For the family of distributions with $k$th moment bounded by 1 for $k \geq 2$, $q(\delta) = k\delta^{1-1/k}$.

Assumption 4.3 expresses the conditional unimodality in likelihood functions in terms of rewards.

**Assumption 4.3.** Given ground truth $f \in \mathcal{F}$ and $g^1, g^2 \in \mathcal{F}$, for every $\pi \in \Pi$, $\mathbb{E}_{g_\pi^1}[r|o] \leq \mathbb{E}_{g_\pi^2}[r|o] \leq \mathbb{E}_{f_\pi}[r|o]$ or $\mathbb{E}_{g_\pi^1}[r|o] \geq \mathbb{E}_{g_\pi^2}[r|o] \geq \mathbb{E}_{f_\pi}[r|o]$ implies $D_{KL}(f_\pi(\cdot \mid o), g_\pi^2(\cdot \mid o)) \leq D_{KL}(f_\pi(\cdot \mid o), g_\pi^1(\cdot \mid o))$ almost everywhere *(a.e.)*.

Assumptions 4.4 and 4.5 exclude trivial cases where the reward information is not at all needed for the inference of the ground-truth model $f$.

**Assumption 4.4** (Density of $\mathcal{F}_\pi$ for every $\pi \in \Pi$)**.** For every $\pi \in \Pi$, $\{g_\pi \in \mathcal{F}_\pi \mid \mu_{g_\pi} \geq \mu_{f_{\pi_f}}\} \cap \{g_\pi \in \mathcal{F}_\pi \mid |\mathbb{E}_{f_\pi}(r \mid o) - \mathbb{E}_{g_\pi}(r \mid o)| \leq q(\delta) \ a.e.\}$ is nonempty given $\delta > 0$.

Intuitively, $\{g_\pi \in \mathcal{F}_\pi \mid \mu_{g_\pi} \geq \mu_{f_{\pi_f}}\}$ is the set of hypotheses in $\mathcal{F}_\pi$ we need to reject, and $\{g_\pi \in \mathcal{F}_\pi \mid |\mathbb{E}_{f_\pi}(r \mid o) - \mathbb{E}_{g_\pi}(r \mid o)| \leq q(\delta) \ a.e.\}$ is the set of hypothesis we cannot reject under contamination of outcomes from decision $\pi$. Note that $\{g_\pi \in \mathcal{F}_\pi \mid |\mathbb{E}_{f_\pi}(r \mid o) - \mathbb{E}_{g_\pi}(r \mid o)| \leq q(\delta) \ a.e.\} \subseteq \{g_\pi \in \mathcal{F}_\pi \mid |\mu_{g_\pi} - \mu_{f_\pi}| \leq q(\delta)\}$.

**Assumption 4.5.** Let $g_\pi^o$ be the marginal distribution of the observation of $g_\pi \in \mathcal{F}_\pi$. There exists $r_o > 0$ such that for every $\pi \in \Pi$, $|\mu_{f_\pi} - \mu_{g_\pi}| \leq r_o$ implies $f_\pi^o = g_\pi^o$ a.e..

Note that $f_\pi^o = g_\pi^o$ a.e. if and only if $D_{KL}(f_\pi^o \| g_\pi^o) = 0$ holds. If $D_{KL}(f_\pi^o \| g_\pi^o) > 0$, no information on the rewards will be required to reject $g_\pi$ under $f_\pi$, the true hypothesis for the decision $\pi$. On the other hand, in the reinforcement learning problems where reward functions are parametrized independent of the transition model parameters, $r_0$ in the Assumption 4.5 is $+\infty$.

## 4.2 Positive Result: $\mathcal{C}(f) = 0$ for all $f \in \mathcal{F}$ enables Super-Robust Bounded Regret

In the previous section, we saw in Theorem 4.1 that $C(f) = 0$ for $f \in \mathcal{F}$ is required to assure the existence of a delay-robust consistent algorithm. We also saw that $C(f) = 0$ for $f \in \mathcal{F}$ is required to assure sub-logarithmic regret for all possible instances of true $f$ (Theorem 2.3).

Here, we try to answer when it is sufficient to assure best of both worlds, i.e., bounded regret and robustness to any delay-model miss-specification at the same time. Before answering this question, we need to define and explore two new concepts: *cross-informativeness* and *max-contamination*.

### 4.2.1 Cross-informativeness

Recall that we denote by $P_{f,n,A}$ the distribution of the outcomes of algorithm $A$ for the model instance $f \in \mathcal{F}$ by the $n$th round. Let us denote the algorithm that always chooses decision $\pi \in \Pi$ as $\overline{\pi}$. Then Lemma 4.6 motivates the concept of cross-informativeness.

**Lemma 4.6.** *Suppose that $f \in \mathcal{F}$ is the ground-truth model instance. Then $C(f) = 0$ implies that $D_{\mathrm{KL}}\left(P_{f,n,\overline{\pi_f}} \| P_{g,n,\overline{\pi_f}}\right) = \Omega(n)$ holds for $g \in \mathcal{F}(f)^c$.*

*Proof.* Suppose that $\mathcal{C}(f) = 0$. According to equation (3), this implies that there exist $w_{\pi_f}$ and $0 \leq n_0 < \infty$ such that for all $n \geq n_0$, $w_{\pi_f} D_{\mathrm{KL}}(f_{\pi_f} \| g_{\pi_f}) \geq 1, \forall g \in \mathcal{F}(f)^c$. Therefore, $D_{\mathrm{KL}}\left(P_{f,n,\overline{\pi_f}} \| P_{g,n,\overline{\pi_f}}\right) = n D_{KL}(f_{\pi_f} \| g_{\pi_f}) \geq \frac{1}{w_{\pi_f}} n$ holds $\forall g \in \mathcal{F}(f)^c$ for $n \geq n_0$. $\qquad \square$

Lemma 4.6 provides an intuitive and straightforward connection between $C(f) = 0$ and bounded regret: we can reject all the hypotheses that need a rejection to conclude that $f$ is indeed the ground truth hypothesis ($\mathcal{F}(f)^c$), simply by exclusively playing $\pi_f$ forever, while incurring zero regret. Lemma 4.6 shows how informative $\pi_f$ is when the true hypothesis is $f$. When the true hypothesis is not $f$, $\pi_f$ can be arbitrarily uninformative.

The natural question now arises is how much cross-informativeness (informativeness of $\pi_h \in \Pi$ for the ground truth $f \in \mathcal{F}$ when $h \neq f$) is sufficient for us to achieve bounded regret. The key

assumption used in this paper is Assumption 4.7, which is later shown to be satisfied for the linear systems (contextual linear bandits, linear MDP) when the conditions implied by $C(f) = 0$ for $f \in \mathcal{F}$ are satisfied (Section 5).

**Assumption 4.7** (Cross-informativeness). Suppose that $f \in \mathcal{F}$ is the ground-truth model. Then for any $g, h \in \mathcal{F}$, $D_{\mathrm{KL}} \left( P_{f,n,\overline{\pi_h}} \| P_{g,n,\overline{\pi_h}} \right) = \omega(\ln n)$ holds.

Note that the cross-informativeness lower bound rate of $\omega(\ln n)$ in Assumption 4.7 is a much weaker rate than the lower bound rate $\Omega(n)$ in Lemma 4.6.

### 4.2.2 Max-contamination

Whatever true delay distribution the reward delays follow, the maximum number of reward arrivals from $\pi$ by the round $k$ is $N_\pi(k)$, the total number of $\pi$ decisions by the round $k$. The *max-contamination* of the decision $\pi' \in \Pi$ at round $k$ is defined as $\delta_{\pi'}^{\max}(k) := \min\left( \frac{\sum_{\pi \in \Pi \setminus \pi'} N_\pi(k)}{\widetilde{N}(k)}, 1 \right)$,

where $\widetilde{N}(k)$ stands for the total number of reward arrivals by round $k$. Note that the contamination of reward arrival at $k$ is bounded by the max-contamination $\delta_{\pi'}^{\max}(k)$, as the delay distributions of decisions are stationary, i.e., they do not change over time.

### 4.2.3 Algorithm Simply-Test-To-Commit (ST2C)

**Assumption 4.8.** For all $f, g \in \mathcal{F}$, $\pi \in \Pi$, and $o \in \mathcal{O}$, $\left| \ln \frac{f_\pi}{g_\pi} \right| < c$ for some $c > 0$. This implies $D_{KL}(f_\pi(\cdot \mid o) \| g_\pi(\cdot \mid o)) < \infty$.

Assumption 4.8 excludes trivially informative cases where $f$ and $g$ are almost immediately distinguished given the observation $o \in \mathcal{O}$. Under Assumption 4.8, we can well-define $\beta := \left( \sup_{g \in \Pi, \pi \in \Pi, E \in \mathcal{E}_\pi} \frac{\mathrm{d} f_\pi(\cdot \mid o)}{\mathrm{d} g_\pi(\cdot \mid o)}(E) \right)^{-1}$ (where $\mathcal{E}_\pi$ denote the collection of measurable sets for $f_\pi(\cdot \mid o)$ and $g_\pi(\cdot \mid o)$), as Assumption 4.8 holds if and only if the log-likelihood ratio $\ln \frac{f_\pi(\cdot \mid o)}{g_\pi(\cdot \mid o)}$ is well-defined on the support of $g_\pi$ and is finite *a.e.*.

Let $P_{g,k,\overline{\pi}}^c$ indicate the likelihood of $g \in \mathcal{F}$ that is computed as if all reward arrivals by the round $k$ are from the decision $\overline{\pi}$. (Note that this is actually not true, as we allow decision transitions in Algorithm 1.) Note that $\ln \frac{P_{f,k,\overline{\pi}}^c}{P_{g,k,\overline{\pi}}^c} = \sum_{k=1}^n \ln \frac{f_\pi^c(k)}{g_\pi^c(k)}$, where $f_\pi^c(k)$ and $g_\pi^c(k)$ are likelihood of each data assuming that the data is from $\pi$.

We now describe the algorithm Simply-Test-to-Commit (ST2C) as the Algorithm 1 below.

---

**Algorithm 1** Simply-Test-to-Commit (ST2C) Algorithm

---

1: Choose any $h \in \mathcal{F}$, set $\widehat{f} = h$
2: **for** $n = 1, 2, \ldots$ **do**
3:     Choose $\pi_{\widehat{f}}$ as the decision at period $n$
4:     Observe $o_n$ and $R_n$, newly compute $\delta_{\pi_{\widehat{f}}}^{\max}(n)$
5:     **if** $\mathcal{F}_n := \{ g \in \mathcal{F} \mid \sum_{k=1}^n \ln \frac{g_{\pi_{\widehat{f}}}^c(k)}{\widehat{f}_{\pi_{\widehat{f}}}^c(k)} \geq 2 \ln n + \sum_{k=1}^n \frac{2}{\sqrt{\beta}} \delta_{\pi_{\widehat{f}}}^{\max}(k) \} \neq \emptyset$ **then**
6:         Choose any $g \in \mathcal{F}_n$
7:         Set $\widehat{f} = g$
8:     **end if**
9: **end for**

---

### 4.2.4 Analysis of algorithm ST2C

Later in Section 5, we will show that $\mathcal{C}(f) = 0$ for all $f \in \mathcal{F}$, which is a design feature of a learning problem we decide a priori, is sufficient to show that Assumption 4.7 indeed holds for some representative linear problems. In this section, we show that Assumption 4.7 (combined with a technical minor Assumption 4.8) is sufficient to allow Algorithm 1 to achieve bounded regret without any knowledge of delay distribution model.

The following Lemma 4.9 shows that $\hat{f}$ stays at incorrect instances for only finite time, except for the periods $\hat{f}$ arrives at incorrect instances.

**Lemma 4.9.** *Under Assumption 4.7 and 4.8, total number of periods $\hat{f}$ stays in $\mathcal{F} \setminus f^\star$ is finite in expectation, except for the periods wrong transition (transition to $\mathcal{F} \setminus f^\star$) happens.*

Lemma 4.10 shows that the total number of wrong transitions from the correct inferences is finite in expectation.

**Lemma 4.10.** *Under Assumption 4.8, the number of rounds Algorithm 1 satisfies the event $\{\widehat{f} = f^\star\} \cap \{\exists g \in \mathcal{F}(f^\star)^c \text{ s.t. } \sum_{k=1}^n \ln \frac{g_{\pi_{\hat{f}}}^c(k)}{f_{\pi_{\hat{f}}}^c(k)} \geq 2 \ln k + \sum_{k=1}^n \frac{2}{\sqrt{\beta}} \delta_{\pi_{\hat{f}}}^{\max}(k)\}$ holds is finite in expectation.*

**Theorem 4.11.** *Under Assumption 4.7 and 4.8, the algorithm ST2C (Algorithm 1), which does not require any knowledge of the delay distribution model, achieves a bounded regret $\Delta(1 + 5\frac{4c^4 e^{-2}}{\mathcal{W}(2c^2)^2})\frac{\pi^2}{6}$, where $\mathcal{W}$ is the principal branch Lambert W function [27], $\Delta$ is the maximum per-period mean reward difference among decisions, and $c$ is from Assumption 4.8.*

*Proof.* Combining Lemmas 4.9 and 4.10, we can conclude that $\widehat{f} \notin \mathcal{F}(f^\star)$ holds only for a finite number of rounds in expectation. That is, regret is bounded in expectation. For detailed derivation of the bound, see Appendix C.3. □

## 5 Equivalence of bounded regret and delay robustness in linear systems

As discussed in Section 4.2, satisfying the cross-informativeness condition introduced in Assumption 4.7 is the key assumption that enables Algorithm 1 to achieve bounded regret with super-robustness to delay. In this section, we show that linear learning problems such as contextual linear bandit and reinforcement learning (RL) with linear MDP indeed satisfies the cross-informativeness condition if $\mathcal{C}(f) = 0$ for $f \in \mathcal{F}$. That is, for those problems, the condition '$\mathcal{C}(f) = 0$ for $f \in \mathcal{F}$' is not only necessary (Section 4.1), but also sufficient for achieving bounded regret under any level of delay model misspecification. In other words, we can conclude that achieving bounded regret is equivalent to achieving *any* level of delay robustness for such linear problems discussed in Section 5.1 and 5.2.

### 5.1 Contextual linear bandit problem

Hao, Lattimore, and Szepesvari [10] was the first to characterize when $\mathcal{C}(f) = 0$ holds for contextual linear bandit problems. In this paper, we follow the notations and settings of [10] as follows: Let's consider the stochastic $M$-armed contextual linear bandit with a horizon of $n$ rounds with $M$ arms and a finite $A$-size set of $k$-dimensional possible contexts $\mathcal{X} = \{\mathbf{x}_j\}_{j \in [A]}$. At each round, a context is sampled according to the unknown distribution $p$ over $\mathcal{X}$ and then observed. Every time a context is sampled, an arm choice (a decision in MDSO framework) happens. When the sampled context is $\mathbf{x}_j$ and its chosen arm is $m$, we receive $\phi_m(\mathbf{x}_j)'\theta + \epsilon$, where $\{\phi_m : \mathbb{R}^k \mapsto \mathbb{R}^d\}_{m \in [M]}$ are linear representation functions that are assumed to be *precisely known*, $\theta$ is a parameter vector of dimension $d$ that is shared across the arms, and $\epsilon$ is an i.i.d. random noise that follows a sub-Gaussian distribution with variance proxy $\sigma^2$.

Let $\Theta$ be the set of all parameter vectors, and let $\theta^\star \in \Theta$ be the unknown true parameter. Suppose that $\mathcal{C}(\theta) = 0$ for $\theta \in \Theta$. Let $m_{j\theta}$ be an optimal arm for context $j \in [A]$ when the true parameter is $\theta$, i.e., $m_{j\theta} \in \text{argmax}_{m \in [M]} \phi_m(\mathbf{x}_j)'\theta$. The following Theorem 5.1 characterizes previous results on when bounded regret can be achieved when there are no reward delays.

**Theorem 5.1** ([10, 28]). *Given linear contextual bandit setting described above, when there are no reward delays, bounded regret can be a priori guaranteed to be achieved if and only if $\{\phi_{m_{j\theta}}(\mathbf{x}_j) \mid j \in A\}$ spans $\mathbb{R}^d$ for all $\theta \in \Theta$.*

Note that the condition that '$\{\phi_{m_{j\theta}}(\mathbf{x}_j) \mid j \in A\}$ spans $\mathbb{R}^d$ for all $\theta \in \Theta$' in Theorem 5.1 is easily satisfied a priori when we are given rich enough context set [29]. How does this easily satisfied condition work when there are anonymous delayed rewards? The following theorem of ours shows that '$\{\phi_{m_{j\theta}}(\mathbf{x}_j) \mid j \in A\}$ spans $\mathbb{R}^d$ for all $\theta \in \Theta$' implies that the Assumption 4.7 is satisfied.

**Theorem 5.2.** *Given contextual linear bandit setting described above, Assumption 4.7 $(D_{\text{KL}}(P_{\theta^\star,n,\overline{\pi_\theta}} \| P_{\theta',n,\overline{\pi_\theta}}) = \Omega(n)$ holds for $\theta' \in \mathcal{F}(\theta^\star)^c)$ is satisfied if $\{\phi_{m_{j\theta}}(\mathbf{x}_j) \mid j \in A\}$ spans $\mathbb{R}^d$ for all $\theta \in \Theta$.*

See Appendix D for the proof of Theorem 5.2. As Assumption 4.8 trivially holds for linear problems [2], Theorem 4.11 (which requires Assumption 4.7 and 4.8 to hold) conclude that our Algorithm 1 achieves bounded regret without any knowledge of delay distribution model if $\{\phi_{m_{j\theta}}(\mathbf{x}_j) \mid j \in A\}$ spans $\mathbb{R}^d$ for all $\theta \in \Theta$ (which is easily satisfied when the context space is rich enough [29]).

For contextual linear bandit setting described above, it has been shown that '$\{\phi_{m_{j\theta}}(\mathbf{x}_j) \mid j \in A\}$ spans $\mathbb{R}^d$ for all $\theta \in \Theta$' is equivalent to '$\mathcal{C}(\theta) = 0$ for all $\theta \in \Theta$' [26, 2, 1]. Since $\mathcal{C}(\theta) = 0$ for all $\theta \in \Theta$ is a necessary condition for a-priori assurance of any-level robustness of a consistent algorithm (Theorem 4.1), we have the following Corollary 5.3, which is a reminiscent of Theorem 5.1 above.

**Corollary 5.3.** *Given contextual linear bandit setting described above, under any anonymous delayed rewards, bounded regret can be a priori guaranteed to be achieved if and only if $\{\phi_{m_{j\theta}}(\mathbf{x}_j) \mid j \in A\}$ spans $\mathbb{R}^d$ for all $\theta \in \Theta$.*

Note that Corollary 5.3 strongly motivates the practical usefulness of bounded regret algorithm design in contextual linear bandit problems, as it is also a necessary and sufficient condition for achieving any level of delay robustness. This condition is indeed not hard to satisfy in the real world; for example, in Spotify, million daily users can be considered a rich enough context for exploring 60,000 new songs uploaded daily [29].

## 5.2 Reinforcement learning with Linear MDP

Papini et al. [11] was the first to characterize the condition for achieving bounded regret for some popular classes of reinforcement learning with episodic Linear Markov Decision Process (MDP). In this paper, we follow the notations of [11], which are as follows: we are given a time-inhomogenous MDP $M = \left(\mathcal{S}, \mathcal{A}, H, \{r_h\}_{h=1}^H, P, \mu\right)$, where $\mathcal{S}$ is finite state space, $\mathcal{A}$ is a finite action space, $H$ is the length of each episode, $\{r_h\}$ are the reward functions where $r_h(s,a)$ the expected reward of a pair $(s,a) \in \mathcal{S} \times \mathcal{A}$ at time-step $h$, $P := \{p_h\}$ are the transition kernels, and $\mu$ is the initial state distribution. A policy $\pi = (\pi_1, \ldots, \pi_H) \in \Pi$ is a sequence of per-time-step policies $\pi_h : \mathcal{S} \to \mathcal{A}$. For every $h \in [H] := \{1, \ldots, H\}$, we define the state-action value function of a policy $\pi$ as $Q_h^\pi(s,a) = r_h(s,a) + \mathbb{E}_\pi\left[\sum_{i=h+1}^H r_i(s_i, a_i)\right]$ and $Q_h^{\pi^\star}(s,a) := Q_h^\star(s,a) = \sup_\pi Q_h^\pi(s,a) = L_h Q_{h+1}^\star(s,a)$ where $L_h Q_{h+1}^\star(s,a) := r_h(s,a) + \mathbb{E}_{s' \sim p_h(s,a)}\left[\max_{a'} Q_{h+1}^\star(s', a')\right]$.

As in [11], we focus on episodic Linear MDP setting with Bellman closure [30], which is more general than many popular Linear MDP settings such as low-rank Linear MDPs [30].

**Definition 5.4** (Linear MDP with Bellman closure (completeness)[30]). Suppose that we are given a feature map $\phi_h : \mathcal{S} \times \mathcal{A} \to \mathbb{R}^{d_h}$, possibly different at any $h \in [H]$, mapping state-action pair $(s,a)$ into a $d_h$-dimensional vector $\phi_h(s,a)$. For the set of bounded value function $\mathcal{Q}_h = \{Q_h \mid \theta_h \in \Theta_h : Q_h(s,a) = \phi_h(s,a)^\top \theta_h, \forall (s,a)\}$ and the associated parameter space $\Theta_h = \{\theta_h \in \mathbb{R}^d : |\phi_h(s,a)^\top \theta_h| \leq D\}$. An MDP is said to satisfy zero Inherent Bellman Error (IBE) (or satisfy Bellman closure) if $\forall h \in [H]$, $\sup_{Q_{h+1} \in \mathcal{Q}_{h+1}} \inf_{Q_h \in \mathcal{Q}_h} \|Q_h - L_h Q_{h+1}\|_\infty = 0$ holds.

The following Theorem 5.5 summarizes a previous result that characterizes when bounded regret can be achieved when there are no reward delays.

**Theorem 5.5** (Papini et al. [11]). *Denote the optimal policy as $\pi^\star$ and $\phi_h^\star(s) := \phi_h(s, \pi_h^\star(s))$. In Linear MDPs satisfying Bellman closure, the condition that 'span$\{\phi_h^\star(s) \mid \forall s \in \mathcal{S}, \pi^\star$ visits $s$ at $h$ with positive probability$\} = \mathbb{R}^d$ for all $h \in [H]$' is sufficient for achieving bounded regret in high probability when there is no unknown reward delay.*

**Theorem 5.6.** *Given episodic Linear MDP with Bellman closure described above, Assumption 4.7 $(D_{\text{KL}}(P_{\theta^\star,n,\overline{\pi_\theta}} \| P_{\theta',n,\overline{\pi_\theta}}) = \Omega(n)$ holds for $\theta' \in \mathcal{F}(\theta^\star)^c)$ is satisfied if span$\{\phi_h^\star(s) \mid \forall s \in \mathcal{S}, \pi^\star$ visits $s$ at $h$ with positive probability$\} = \mathbb{R}^d$ for all $h \in [H]$.*

Again, as Assumption 4.8 trivially holds for linear problems [2], Theorem 4.11 (which requires Assumption 4.7 and 4.8 to hold) conclude that our Algorithm 1 achieves bounded regret without

any knowledge of delay distribution model if $\mathrm{span}\{\phi_h^\star(s) \mid \forall s \in \mathcal{S}, \pi^\star$ visits $s$ at $h$ with positive probability$\} = \mathbb{R}^d$ for all $h \in [H]$. Therefore, we have the following Corollary 5.7.

**Corollary 5.7.** *Given episodic Linear MDP with Bellman closure setting described above, under any anonymous delayed rewards, bounded regret can be a priori guaranteed to be achieved if* $\mathrm{span}\{\phi_h^\star(s) \mid \forall s \in \mathcal{S}, \pi^\star$ *visits* $s$ *at* $h$ *with positive probability}* $= \mathbb{R}^d$ *for all* $h \in [H]$.

Again, note that Corollary 5.7 strongly motivates the practical usefulness of bounded regret algorithm design, as the sufficient condition for it is also sufficient for achieving any level of delay robustness.

## 6 Conclusions

In this paper, we characterize the link between consistent algorithms and delay robustness in inter-active decision-making. The first main result states that, for consistent algorithms, the necessary condition for achieving any (small) level of robustness against delay model misspecifications is also sufficient for achieving bounded regret. Viewed from another perspective, this result urges us to revisit the practicality of the instance-dependent regret minimizing algorithm design regime [9, 1] for real-world problems with anonymous delayed rewards. The second main result states vice versa for linear problems, showing that the well-known necessary (and sufficient) condition for bounded regret is also sufficient for designing a consistent algorithm that achieves any (large) level of robustness against delay model misspecifications and bounded regret at the same time. An interesting future research direction raised by our paper is whether it is possible to achieve our second main result without restricting to linear problems.

## Acknowledgements

This material is based upon work partially supported by the US Army Contracting Command under W911NF-22-1- 0151 and USARO under W911NF2120064, the US National Science Foundation under CNS-2328395 and CMMI-2038625, and the US Office of Naval Research under N00014-24-1-2615 and N00014-21-1-2385. This work was also partially conducted with support from the Bertauche Transportation Endowment and the Edna Benson PhD Fellowship.

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

## A  Special Cases of DMSO Framework

- In finite-armed bandit problems, each round is an arm pull. $\Pi$ is the arm space, and $\mathcal{R}$ is the space of rewards from arms. Since there is no observation space $\mathcal{O}$, the model class degenerates to $\mathcal{F} \subseteq (\Pi \mapsto \Delta_{\mathcal{R}})$.

- In contextual bandit problems, each round is an arm pull. $\Pi$ is the set $(\mathcal{X} \mapsto \mathcal{A})$ of all policies, where $\mathcal{X}$ is the context space and the $\mathcal{A}$ is the arm space. The reward space $\mathcal{R}$ is the space of rewards from arms. The observation space $\mathcal{O}$ is $\mathcal{X}$, where the $k$th round's observation $o_k \in \mathcal{O}$ (which results from $\pi_k$) is the $k + 1$th round's context. Since the future context arrival is not affected by previous decisions, the model class degenerates to $\mathcal{F} \subseteq (\Pi \mapsto \Delta_{\mathcal{R}})$.

- In episodic reinforcement learning problems, each round is an episode. $\Pi$ is the set $(\mathcal{S} \mapsto \mathcal{A})$ of all policies, where $\mathcal{S}$ is the space of all possible states and $\mathcal{A}$ is the action space. The reward space $\mathcal{R}$ is the space of value functions at each initial state, and the observation space $\mathcal{O}$ is the set of all possible sequences of action choices, state transitions, and received rewards in one episode. The model class $\mathcal{F}$ is characterized jointly by the initial state distribution and the transition kernel, which are shared across all the episodes.

## B  Proof of Theorem 4.1

Recall that $P_{f,n,A}[\cdot]$ denotes the distribution of outcomes of algorithm $A$ on the true model instance $f$ by the round $n$. We further denote the marginal distribution of $P_{f,n,A}[\cdot]$ in terms of decision $\pi$'s rewards and outcomes by $P_{f,n,A}^{\pi}[\cdot]$.

**Lemma B.1.** *Suppose that the ground-truth model is $f \in \mathcal{F}$. Then a consistent algorithm must satisfy* $(1 + o(1)) \ln n \leq \sum_{\pi \in \Pi} D_{\mathrm{KL}} \left( P_{f,n,A}^{\pi} \| P_{g,n,A}^{\pi} \right)$ *for $g \in \mathcal{F}(f)^c$.*

*Proof.* According to Dong and Ma (2022) [2], any consistent algorithm $A$ must satisfy $(1 + o(1)) \ln n \leq D_{\mathrm{KL}} (P_{f,n,A} \| P_{g,n,A})$ for $g \in \mathcal{F}(f)^c$. Since the terms involving $A$ (the algorithm used to collect the data) cancel out and the outcomes of decisions are independent of each other, $\frac{P_{f,n,A}}{P_{g,n,A}} = \prod_{\pi \in \Pi} \frac{P_{f,n,A}^{\pi}}{P_{g,n,A}^{\pi}}$ holds. Therefore, the condition of Dong and Ma (2022) becomes $(1 + o(1)) \ln n \leq \sum_{\pi \in \Pi} D_{\mathrm{KL}} \left( P_{f,n,A}^{\pi} \| P_{g,n,A}^{\pi} \right)$ for $g \in \mathcal{F}(f)^c$. $\qquad\square$

**Lemma B.2.** *For any $\epsilon > 0$, $\epsilon$-contamination in the delay model of $\pi^{\star}$ makes the rewards of decisions $\pi \neq \pi^{\star}$ suffer $\delta$-contamination for some $\delta > 0$ under consistency.*

See Section B.1 for the proof of Lemma B.2.

Lemma B.3 shows that, under $\delta$-reward contaminations in reward distributions of all $\pi \neq \pi^{\star}$, choosing optimal decision $\pi^{\star}$ alone must be enough to satisfy the condition described in Lemma B.1 and otherwise, we cannot satisfy it.

**Lemma B.3.** *If the rewards of decisions $\pi \in \Pi \setminus \pi_f$ suffer $\delta$-contamination for some $\delta > 0$, consistency of the algorithm requires* $(1 + o(1)) \ln n \leq D_{\mathrm{KL}} \left( P_{f,n}^{\pi_f} \| P_{g,n}^{\pi_f} \right)$ *for all $g \in \mathcal{F}(f)^c$.*

See Section B.2 for the proof of Lemma B.3.

The rest of the proof of Theorem 4.1 is immediate from the derivation of [2]'s Theorem 2.3, which is as follows: from the chain rule of divergence, $D_{\mathrm{KL}} \left( P_{f,n}^{\pi} \| P_{g,n}^{\pi} \right) = \mathbb{E}_{f,n} [N_{\pi}] D_{\mathrm{KL}}(f_{\pi} \| g_{\pi})$ holds for $\pi \in \Pi$. Defining $w_{\pi} := \mathbb{E}_f [N_{\pi}] / ((1 + o(1)) \ln n)$, Lemma B.3 implies that $\mathcal{C}(f) = 0$ by the definition of $\mathcal{C}(f)$ in the equation (3). Since we don't know the ground truth $f$ a priori, designing a learning system that assures the existence of a robust algorithm requires $\mathcal{C}(f) = 0$ for all $f \in \mathcal{F}$.

### B.1  Proof of Lemma B.2

Denote by $N_{\pi}^{[a,b]}$ the random variable that counts the number of decisions of $\pi$ between rounds $a$ and $b$. For consistency, for any small enough $p > 0$, for any $r > 0$, for some $m$, there must exist a constant

$n_{r,p}$ such that for all intervals $[a,b]$ with $b-a \geq n_{r,p}$ and $a,b > m$, $E[N_{\pi^\star}^{[a,b]}] \geq (b-a) - (b-a)^p r$ holds. Recall that we denote by $D_\pi$ the true delay model for the decision $\pi \in \Pi$, and by $\hat{D}_\pi$ the given model for $D_\pi$. Note that $\epsilon$-contamination means we can arbitrarily choose $\{D_\pi\}_{\pi \in \Pi}$ as long as $d_{TV}(D_\pi, \hat{D}_\pi) \leq \epsilon$. Consider the case when $D_{\pi^\star} = \hat{D}_{\pi^\star} + \epsilon D_a - \epsilon D_c$ where $P(D_a = k) = \frac{1}{n_{r,p}}$ for $0 \leq k \leq n_{r,p} - 1$ and 0 for elsewhere, and $D_c$ is an arbitrary distribution. For $\pi \in \Pi \setminus \pi^\star$, consider $D_\pi = \hat{D}_\pi$. Then for $k \geq \max(n_{r,p}, m)$,

$$P(\{\text{A reward arrival at } k \text{ is not from } \pi^\star\})$$
$$= \frac{\sum_{i=1}^k P(\{\pi_i\text{'s reward arrives at } k \text{ and } \pi_i \neq \pi^\star\})}{\sum_{i=1}^k P(\{\pi_i\text{'s reward arrives at } k\})}$$
$$\leq \frac{|\Pi| - 1}{|\Pi| - 1 + \sum_{i=1}^k P(\{\pi_i\text{'s reward arrives at } k \text{ and } \pi_i = \pi^\star\})} \quad (4)$$
$$\leq \frac{|\Pi| - 1}{|\Pi| - 1 + \frac{\epsilon}{n_{r,p}} \sum_{i=k-n_{r,p}}^k \mathbb{E}[1_{\pi_i = \pi^\star}]}$$
$$\leq \frac{|\Pi| - 1}{|\Pi| - 1 + \frac{\epsilon}{n_{r,p}}(n_{r,p} - n_{r,p}^p r)} = \frac{|\Pi| - 1}{|\Pi| - 1 + \epsilon(1 - n_{r,p}^{p-1} r)}$$

where the inequality in the equation (4) follows from the fact that the delay distribution of each decision sums to one. Therefore, $P(\{\text{A reward arrival at } k \text{ is from } \pi^\star\}) > \delta := \frac{\epsilon(1 - n_{r,p}^{p-1} r)}{|\Pi| - 1 + \epsilon(1 - n_{r,p}^{p-1} r)}$.

Denote the rewards distributions associated with $D_{\pi^\star}, \hat{D}_{\pi^\star}, D_a$, and $D_c$ as $R_{\pi^\star}, \hat{R}_{\pi^\star}, R_a$, and $R_c$ each. Then $R_{\pi^\star} = \hat{R}_{\pi^\star} + \epsilon R_a - \epsilon R_c$ must hold, where the mean of $R_{\pi^\star}$ and $R_{\pi^\star}$ are supposed to be the same. Since the choice of $R_a$ can be arbitrary by choosing $R_c$ accordingly, we can conclude that the reward distributions of decisions $\pi \neq \pi^\star$ indeed suffer $\delta$-contamination.

## B.2 Proof of Lemma B.3

Recall that we use $f_\pi$ to refer to an element of $\mathcal{F}_\pi$, while $f_\pi$ also denotes the $\pi$-coordinate of some $f \in \mathcal{F}$. Suppose that the ground-truth model is $f \in \mathcal{F}$. For $g \in \mathcal{F}(f)^c$, a consistent algorithm must satisfy

$$(1 + o(1)) \ln n \leq D_{\text{KL}}(P_{f,n} \| P_{g,n}) = \sum_{\pi \in \Pi} D_{\text{KL}}(P_{f,n}^\pi \| P_{g,n}^\pi). \quad (5)$$

$$= D_{\text{KL}}\left(P_{f,n}^{\pi_f} \| P_{g,n}^{\pi_f}\right) + \sum_{\pi \in \Pi \setminus \pi_f} \mathbb{E}_{f,n}[N_\pi] D_{\text{KL}}(f_\pi(r,o) \| g_\pi(r,o)) \quad (6)$$

$$= D_{\text{KL}}\left(P_{f,n}^{\pi_f} \| P_{g,n}^{\pi_f}\right) + \sum_{\pi \in \Pi \setminus \pi_f} \mathbb{E}_{f,n}[N_\pi]\left(D_{\text{KL}}(f_\pi^o(o) \| g_\pi^o(o)) + \mathbb{E}_{f_\pi}\left[\log \frac{f_\pi(r|o)}{g_\pi(r|o)}\right]\right) \quad (7)$$

Above,

- The inequality in the equation (5) is from Dong and Ma (2022) [2].

- The equality in the equation (5) follows from the fact that algorithm-related terms cancel out.

- The inequality in the equation (6) is from the Divergence decomposition Lemma [31]

- The inequality in the equation (7) follows from the chain rule of KL divergence [32].

Let $\min(q(\delta), r_o) = q'(\delta)$, where $r_o$ is defined as in Assumption 4.5. By Assumption 4.4, for every $\pi \neq \pi_f$, there exists a non-empty set $\mathcal{E}_\pi := \{l_\pi \in \mathcal{F}_\pi \mid |E_{f_\pi}[r|o] - E_{l_\pi}[r|o]| \leq q'(\delta) \ a.e.\} \cap \{\mu_{l_\pi} \geq \mu_{f_{\pi_f}}\}$. Hense we can construct $\mathcal{E}(\pi) := \{l \in \mathcal{F} \mid l_\pi \in \mathcal{E}_\pi, l_{\pi'} = f_{\pi'} \text{ for } \pi' \neq \pi\}$. Note that $\mathcal{E}(\pi) \subseteq \mathcal{F}(f)^c := \{g \in \mathcal{F} \mid \pi_g \neq \pi_f\} = \{g \in \mathcal{F} \mid \exists \pi \in \Pi \ s.t. \ \mu_{g_\pi} \geq \mu_{f_{\pi_f}}\}$. Therefore, for every

$\pi \neq \pi_f$,

$$(1 + o(1)) \ln n \leq D_{\mathrm{KL}}\left(P_{f,n}^{\pi_f} \| P_{g,n}^{\pi_f}\right) + \sum_{\pi \in \Pi \setminus \pi_f} \mathbb{E}_{f,n}\left[N_\pi(n)\right] \mathbb{E}_{f_\pi}\left[\log \frac{f_\pi(r|o)}{g_\pi(r|o)}\right] \text{ for } g \in \mathcal{E}(\pi) \quad (8)$$

$$(\Rightarrow) \ (1 + o(1)) \ln n \leq D_{\mathrm{KL}}\left(P_{f,n}^{\pi_f} \| P_{g,n}^{\pi_f}\right) + \mathbb{E}_{f,n}\left[N_\pi(n)\right] \mathbb{E}_{f_\pi}\left[\log \frac{f_\pi(r|o)}{g_\pi(r|o)}\right] \text{ for } g \in \mathcal{E}(\pi) \quad (9)$$

$$(\Rightarrow) \ (1 + o(1)) \ln n \leq D_{\mathrm{KL}}\left(P_{f,n}^{\pi_f} \| P_{g,n}^{\pi_f}\right) \text{ for } g \in \mathcal{E}(\pi) \quad (10)$$

$$(\Rightarrow) \ (1 + o(1)) \ln n \leq D_{\mathrm{KL}}\left(P_{f,n}^{\pi_f} \| P_{g,n}^{\pi_f}\right) \text{ for } g \in \mathcal{E}'(\pi) \quad (11)$$

$$(\Rightarrow) \ (1 + o(1)) \ln n \leq D_{\mathrm{KL}}\left(P_{f,n}^{\pi_f} \| P_{g,n}^{\pi_f}\right) \text{ for } g \in \{g \in \mathcal{F} \mid \mu_{g_\pi} \geq \mu_{f_{\pi_f}}\}. \quad (12)$$

where $\mathcal{E}'_\pi := \{l_\pi \in \mathcal{F}_\pi \mid \mu_{f_{\pi_f}} \leq \mu_{l_\pi}\}$ and $\mathcal{E}'(\pi) := \{l \in \mathcal{F} \mid l_\pi \in \mathcal{E}'_\pi, l_{\pi'} = f_{\pi'} \text{ for } \pi' \neq \pi\}$. Above,

- Equation (8) follows from Assumption 4.5 and equation (7).
- The logical implication in equation (9) follows from the definition of $\mathcal{E}(\pi)$.
- The logical implication in equation (10) follows from Assumptions 4.2, 4.3 and 4.4:
  $|E_{f_\pi}[r|o] - E_{g_\pi}[r|o]| \leq q'(\delta) \leq q(\delta)$ a.e. implies $d_{TV}(f_\pi(\cdot \mid o), g_\pi(\cdot \mid o)) < \delta$ a.e. from Assumption 4.2; therefore, under some $\delta$-contamination of $f_\pi(\cdot \mid o)$, the contaminated $E_{f_\pi}[r|o]$ can be farther from the true $E_{f_\pi}[r|o]$ than $E_{g_\pi}[r|o]$. Therefore, $D_{\mathrm{KL}}(f_\pi(\cdot \mid o) \| g_\pi(\cdot \mid o)) \leq 0$ a.e. due to Assumption 4.3, and so $\mathbb{E}_{f_\pi}\left[\log \frac{f_\pi(r|o)}{g_\pi(r|o)}\right] = \mathbb{E}_{f_\pi^o}\left[\mathbb{E}_{f_\pi(r|o)}\left[\log \frac{f_\pi(r|o)}{g_\pi(r|o)}\right]\right] = \mathbb{E}_{f_\pi^o}\left[D_{\mathrm{KL}}(f_\pi(\cdot \mid o) \| g_\pi(\cdot \mid o))\right] \leq 0.$
- The logical implication in equation (11) follows from Assumption 4.3:
  Define $\hat{\mathcal{E}}_\pi := \{|\mu_{f_\pi} - \mu_{g_\pi}| \leq q'(\delta)\} \cap \{\mu_{g_\pi} \geq \mu_{f_{\mu_f}}\}$ and $\hat{\mathcal{E}}(\pi) := \{l \in \mathcal{F} \mid l_\pi \in \hat{\mathcal{E}}_\pi, l_{\pi'} = f_{\pi'} \text{ for } \pi' \neq \pi\}$. Note that $\hat{\mathcal{E}}(\pi) \subseteq \mathcal{E}(\pi)$. Then for $g' \in \mathcal{E}'(\pi) \setminus \hat{\mathcal{E}}(\pi)$ and $g \in \hat{\mathcal{E}}(\pi)$ with $(\mu_{g_\pi} - \mu_{f_\pi})(\mu_{g'_\pi} - \mu_{f_\pi}) \geq 0$, $D_{\mathrm{KL}}\left(P_{f,n}^\pi \| P_{g,n}^\pi\right) \leq D_{\mathrm{KL}}\left(P_{f,n}^\pi \| P_{g',n}^\pi\right)$ due to the monotonicity assumption of Assumption 4.3.
- The logical implication in equation (12) follows from the fact that any element in $\{g \in \mathcal{F} \mid \mu_{g_\pi} \geq \mu_{f_{\pi_f}}\}$ has an element in $\mathcal{E}'$ that is strictly closer to $f$.

Since $\mathcal{F}(f)^c := \{g \in \mathcal{F} \mid \pi_g \neq \pi_f\} = \{g \in \mathcal{F} \mid \exists \pi \in \Pi \ s.t. \ \mu_{g_\pi} \geq \mu_{f_{\pi_f}}\}$, we immediately get $(1 + o(1)) \ln n \leq D_{\mathrm{KL}}\left(P_{f,n}^{\pi_f} \| P_{g,n}^{\pi_f}\right)$ for $g \in \mathcal{F}(f)^c$.

# C  Proof of Lemma 4.9, Lemma 4.10 and Theorem 4.11

## C.1  Proof of Lemma 4.9

**Lemma C.1** (Upper bounding lemma). $k = \frac{4c^4 e^{-2}}{\mathcal{W}(2c^2)^2}$ *satisfies* $e^{-\frac{n(\ln n)^2}{2c^2}} \le k\frac{1}{n^2}$ *for all* $n \ge 1$, *where* $\mathcal{W}$ *is the principal branch Lambert W function [27].*

*Proof.*

$$y = x^2 \cdot e^{-\frac{x(\ln x)^2}{2c^2}}$$

$$= x^2 \cdot e^{-f(x)} \quad \text{(by setting } f(x) = \frac{x(\ln x)^2}{2c^2}$$

$$\frac{dy}{dx} = \frac{d}{dx}\left(x^2\right) \cdot e^{-f(x)} + x^2 \cdot \frac{d}{dx}\left(e^{-f(x)}\right)$$

$$= 2x \cdot e^{-f(x)} + x^2 \cdot e^{-f(x)}\left(-\left(\frac{(\ln x)^2}{2c^2} + \frac{\ln x}{c^2}\right)\right)$$

$$\left(\because \frac{d}{dx}(e^{-f(x)}) = e^{-f(x)} \cdot (-f'(x)) \text{ and}\right.$$

$$f'(x) = \frac{d}{dx}\left(\frac{x(\ln x)^2}{2c^2}\right) = \frac{1}{2c^2}\left((\ln x)^2 + 2x\ln x \cdot \frac{1}{x}\right) = \frac{(\ln x)^2}{2c^2} + \frac{\ln x}{c^2})$$

$$= 2xe^{-f(x)} - x^2 e^{-f(x)}\left(\frac{(\ln x)^2}{2c^2} + \frac{\ln x}{c^2}\right)$$

$$= e^{-f(x)}\left(2x - x^2\left(\frac{(\ln x)^2}{2c^2} + \frac{\ln x}{c^2}\right)\right)$$

Set the derivative to zero to find the critical points:

$$2x - x^2\left(\frac{(\ln x)^2}{2c^2} + \frac{\ln x}{c^2}\right) = 0$$

$$2 = x\left(\frac{(\ln x)^2}{2c^2} + \frac{\ln x}{c^2}\right) \text{ if } x > 0$$

$$4c^2 = x(\ln x)^2 + 2x\ln x$$

This is the uni-modal function with a maximum, as the derivatives are positive on the left side of the critical point and negative on the right side of the critical point. Also, note that $x_{\max}\ln x_{\max} \le 2c^2$, as $x(\ln x)^2 \ge 0$. This implies that $x_{\max} \le \frac{2c^2}{\mathcal{W}(2c^2)}$, where $\mathcal{W}$ denote the principal branch Lambert $W$ function. Now note that

$$y_{\max} = x_{\max}^2 \cdot e^{-\frac{x_{\max}(\ln x_{\max})^2}{2c^2}} \tag{13}$$

$$= x_{\max}^2 \cdot e^{-\left(2 + \frac{x_{\max}}{c^2}\ln x_{\max}\right)} \tag{14}$$

$$\le e^{-2} \cdot x_{\max}^2 \tag{15}$$

$$\le e^{-2}\left(\frac{2c^2}{\mathcal{W}(2c^2)}\right)^2 = \frac{4c^4 e^{-2}}{\mathcal{W}(2c^2)^2} \tag{16}$$

Therefore, $k = \frac{4c^4 e^{-2}}{\mathcal{W}(2c^2)^2}$ satisfies $e^{-\frac{n(\ln n)^2}{2c^2}} \le k\frac{1}{n^2}$ for all $n \ge 1$.

$\square$

*Proof of Lemma 4.9.* Let $f$ be the ground truth model. After each of $\hat{f}$'s transition to $g \in \mathcal{F}(f)^c$, at period $n$,

$$P(\{\sum_{k=1}^n \ln \frac{f_{\pi_g}^c(k)}{g_{\pi_g}^c(k)} \leq 2\ln n + \sum_{k=1}^n \frac{2}{\sqrt{\beta}} \delta_{\pi_g}^{\max}(k)\})$$

$$= P(\{\sum_{k=1}^n (\ln \frac{f_{\pi_g}}{g_{\pi_g}} + \ln \frac{f_{\pi_g}^c(k)}{f_{\pi_g}} + \ln \frac{g_{\pi_g}}{g_{\pi_g}^c(k)}) \leq 2\ln n + \sum_{k=1}^n \frac{2}{\sqrt{\beta}} \delta_{\pi_g}^{\max}(k)\})$$

$$\leq P(\{\sum_{k=1}^n (\ln \frac{f_{\pi_g}}{g_{\pi_g}} + \ln \frac{f_{\pi_g}^c(k)}{f_{\pi_g}} + \ln \frac{g_{\pi_g}}{g_{\pi_g}^c(k)}) \leq 2\ln n + \frac{2C}{\sqrt{\beta}} \ln n\}) \text{ for some } C \tag{17}$$

$$\leq P(\{\sum_{k=1}^n \left(\ln \frac{f_{\pi_g}}{g_{\pi_g}} - D_{KL}(f_{\pi_g}, g_{\pi_g})\right) + \sum_{k=1}^n \left(\ln \frac{f_{\pi_g}^c(k)}{f_{\pi_g}}\right) + \sum_{k=1}^n \left(\ln \frac{g_{\pi_g}}{g_{\pi_g}^c(k)}\right)$$
$$\leq -\ln n\}) \text{ for } n \geq n_0 \text{ for some } n_0 < \infty \tag{18}$$

$$\leq P(\{\sum_{k=1}^n \left(\ln \frac{f_{\pi_g}}{g_{\pi_g}} - D_{KL}(f_{\pi_g}, g_{\pi_g})\right) + \sum_{k=1}^n \left(\ln \frac{f_{\pi_g}^c(k)}{f_{\pi_g}} - D_{KL}(f_{\pi_g}, f_{\pi_g}^c(k))\right)$$
$$+ \sum_{k=1}^n \left(\ln \frac{g_{\pi_g}}{g_{\pi_g}^c(k)} - D_{KL}(g_{\pi_g}, g_{\pi_g}^c(k))\right) \leq -\ln n\}) \text{ for } n \geq n_0 \tag{19}$$

$$\leq P(\{\sum_{k=1}^n \left(\ln \frac{f_{\pi_g}}{g_{\pi_g}} - D_{KL}(f_{\pi_g}, g_{\pi_g})\right) \geq -\ln n, \sum_{k=1}^n \left(\ln \frac{f_{\pi_g}^c(k)}{f_{\pi_g}} - D_{KL}(f_{\pi_g}, f_{\pi_g}^c(k))\right)$$
$$\geq -\ln n, \sum_{k=1}^n \left(\ln \frac{g_{\pi_g}}{g_{\pi_g}^c(k)} - D_{KL}(g_{\pi_g}, g_{\pi_g}^c(k))\right) \geq -\ln n\}^{\mathbf{c}}) \text{ for } n \geq n_0$$

$$= P(\{\sum_{k=1}^n \left(\ln \frac{f_{\pi_g}}{g_{\pi_g}} - D_{KL}(f_{\pi_g}, g_{\pi_g})\right) \leq -\ln n\} \cup \{\sum_{k=1}^n \left(\ln \frac{f_{\pi_g}^c(k)}{f_{\pi_g}} - D_{KL}(f_{\pi_g}, f_{\pi_g}^c(k))\right)$$
$$\leq -\ln n\} \cup \{\sum_{k=1}^n \left(\ln \frac{g_{\pi_g}}{g_{\pi_g}^c(k)} - D_{KL}(g_{\pi_g}, g_{\pi_g}^c(k))\right) \leq -\ln n\}) \text{ for } n \geq n_0$$

$$\leq 3e^{-\frac{n(\ln n)^2}{2c^2}} \tag{20}$$

$$\leq \frac{12c^4 e^{-2}}{\mathcal{W}(2c^2)^2} \frac{1}{n^2} \tag{21}$$

Above,

- Equation 17 follows from the fact that $\delta_{\pi_g}^{\max}(k)$ decreases with the rate $1/n$.

- Equation 18 follows from the fact that $nD_{KL}(f_{\pi_g}, g_{\pi_g}) = D_{KL}(P_{f,n,\overline{\pi_g}} \| P_{g,n,\overline{\pi_g}}) = \omega(\ln n)$ from the Assumption 4.7.

- Equation 19 follows from the fact that substracting positive value on the left does not change the inequality.

- Equation 20 follows from the fact that the log-likelihood ratios are bounded by constant $c$ due to Assumption 4.8, and thus sub-gaussian random variables with $\sigma^2 = \frac{c^2}{4}$.

- Equation 21 is from Lemma C.1.

Therefore, after each time a bad transition to $g \in \mathcal{F}(f)^c$ happens, the event $\{\sum_{k=1}^n \ln \frac{f_{\pi_g}^c(k)}{g_{\pi_g}^c(k)} \leq 2\ln n + \sum_{k=1}^n \frac{2}{\sqrt{\beta}} \delta_{\pi_g}^{\max}(k)\}$ happens only finite many times (more precisely, smaller than $3 \cdot \frac{4c^4 e^{-2}}{\mathcal{W}(2c^2)^2} \cdot \frac{\pi^2}{6} = \frac{2c^4 e^{-2} \pi^2}{\mathcal{W}(2c^2)^2}$) in expectation by the Borel-Cantelli lemma, which implies that the inference will arrive at the correct instance within finite expected time. $\qquad\square$

## C.2 Proof of Lemma 4.10

*Proof of Lemma 4.10.* Suppose that $f$ is the ground truth model. For any $g \in \mathcal{F} \setminus f^\star$, when $\hat{f} = f$,

$$P(\{\sum_{k=1}^{n} \ln \frac{g_{\pi_f}^c(k)}{f_{\pi_f}^c(k)} \geq 2 \ln n + \sum_{k=1}^{n} \frac{2}{\sqrt{\beta}} \delta_{\pi_f}^{\max}(k)\})$$

$$= P(\{\sum_{k=1}^{n} (\ln \frac{g_{\pi_f}}{f_{\pi_f}} + \ln \frac{f_{\pi_f}}{f_{\pi_f}^c(k)} + \ln \frac{g_{\pi_f}^c(k)}{g_{\pi_f}}) \geq 2 \ln n + \sum_{k=1}^{n} \frac{2}{\sqrt{\beta}} \delta_{\pi_f}^{\max}(k)\})$$

$$\leq P(\{\sum_{k=1}^{n} \left( \ln \frac{g_{\pi_f}}{f_{\pi_f}} \right) + \sum_{k=1}^{n} \left( \ln \frac{f_{\pi_f}}{f_{\pi_f}^c(k)} - D_{KL}(f_{\pi_f}, f_{\pi_f}^c(k)) \right)$$

$$+ \sum_{k=1}^{n} \left( \ln \frac{g_{\pi_f}^c(k)}{g_{\pi_f}} - D_{KL}(g_{\pi_f}^c(k), g_{\pi_f}) \right) \geq 2 \ln n\}) \tag{22}$$

$$\leq P(\{\sum_{k=1}^{n} \left( \ln \frac{g_{\pi_f}}{f_{\pi_f}} \right) \leq 2 \ln n, \sum_{k=1}^{n} \left( \ln \frac{f_{\pi_f}}{f_{\pi_f}^c(k)} - D_{KL}(f_{\pi_f}, f_{\pi_f}^c(k)) \right) \leq 2 \ln n,$$

$$, \sum_{k=1}^{n} \left( \ln \frac{g_{\pi_f}^c(k)}{g_{\pi_f}} - D_{KL}(g_{\pi_f}^c(k), g_{\pi_f}) \right) \leq 2 \ln n\}^{\mathbf{c}})$$

$$\leq P(\{\sum_{k=1}^{n} \left( \ln \frac{g_{\pi_f}}{f_{\pi_f}} \right) \geq 2 \ln n\} \cup \{\sum_{k=1}^{n} \left( \ln \frac{f_{\pi_f}}{f_{\pi_f}^c(k)} - D_{KL}(f_{\pi_f}, f_{\pi_f}^c(k)) \right) \geq 2 \ln n\}$$

$$, \cup \{\sum_{k=1}^{n} \left( \ln \frac{g_{\pi_f}^c(k)}{g_{\pi_f}} - D_{KL}(g_{\pi_f}^c(k), g_{\pi_f}) \right) \geq 2 \ln n\})$$

$$\leq \frac{1}{n^2} + 2 \frac{4c^4 e^{-2}}{\mathcal{W}(2c^2)^2} \frac{1}{n^2} \tag{23}$$

Above,

- Equation 22 follows from the reverse Pinsker's inequality [33] (total variation distance smaller than $\delta$ implies KL divergence smaller than $\frac{1}{\sqrt{\beta}}\delta$)

- Equation 23 follows from Lemma 4.3 of Dong and Ma (2022) [2], which says that $P_Q \left( \left\{ \sum_{i=1}^{m} \ln \frac{P_i}{Q_i} \geq c \right\} \right) \leq \exp(-c)$, and the fact that the log-likelihood ratios are bounded due to Assumption 4.8, and thus sub-gaussian random variables.

Therefore, the event holds in total only for finite rounds of $k$ in expectation (more precisely, bounded by $(1 + \frac{8c^4 e^{-2}}{\mathcal{W}(2c^2)^2}) \frac{\pi^2}{6}$) by the Borel-Cantelli lemma. □

## C.3 Proof of Theorem 4.11

Combining Lemmas 4.9 and 4.10, we can conclude that $\hat{f} \notin \mathcal{F}(f^\star)$ holds only for a finite number of rounds in expectation. That is, regret is bounded in expectation with value $\Delta(1 + 5 \frac{4c^4 e^{-2}}{\mathcal{W}(2c^2)^2}) \frac{\pi^2}{6}$.

## D  Proof of Theorem 5.2 (Linear contextual bandit case)

Let $\Theta$ be the set of all parameters, and let $\theta^\star \in \Theta$ be the unknown true parameter. Suppose that $\mathcal{C}(\theta) = 0$ for $\theta \in \Theta$. By Theorem 2.3 and Theorem 5.1, $\left\{ \phi_{m_{j\theta}}(x_j) \mid j \in A \right\}$ spans $\mathbb{R}^d$ for $\theta \in \Theta$. Denote $T_{\mathbf{x}}(n)$ be the number of arrivals of context $\mathbf{x} \in \mathcal{X}$.

Then for any $\tilde{\theta} \in \Theta \setminus \{\theta^\star\}$,

$$D_{\mathrm{KL}}\left( P_{\theta^\star, n, \overline{\pi_\theta}} \| P_{\tilde{\theta}, n, \overline{\pi_\theta}} \right) = \frac{1}{2} \sum_{x \in \mathcal{A}} \mathbb{E}\left[ T_x(n) \right] \langle x, \theta^\star - \widetilde{\theta} \rangle^2 \tag{24}$$

$$= \frac{1}{2} (\theta^\star - \widetilde{\theta})^\top \mathbb{E}\left[ \sum_{x \in \mathcal{A}} T_x(n) x x^\top \right] (\theta^\star - \widetilde{\theta})$$

$$= \frac{1}{2} (\theta^\star - \widetilde{\theta})^\top n \mathbb{E}\left[ \sum_{x \in \mathcal{A}} \frac{T_x(n)}{n} x x^\top \right] (\theta^\star - \widetilde{\theta})$$

$$\geq \frac{1}{2} \| \theta^\star - \widetilde{\theta} \|^2 n \lambda_{\min} \tag{25}$$

$$= \Omega(n) \tag{26}$$

Above,

- The equality in equation (24) is from the divergence decomposition lemma Lattimore and Szepesvári [31]

- $\lambda_{\min}$ of equation (25) denotes the smallest eigenvalue for $\mathbb{E}_{\mathbf{x}_j \sim p}\left[ \phi_{m_{j\theta}}(\mathbf{x}_j) \phi_{m_{j\theta}}(\mathbf{x}_j)^\top \right]$

- The inequality of equation (25) is from the fact that $\frac{x^T A x}{x^T x}$ is larger than the smallest eigenvalue of $A$.

- The equality of equation (26) comes from the fact that $\lambda_{\min} > 0$ is equivalent to $\left\{ \phi_{m_{j\theta}}(x_j) \mid j \in A \right\}$ spanning $\mathbb{R}^d$ [28].

## E  Proof of Theorem 5.6 (Reinforcement learning with Linear MDP case)

It is straightforward that the proof of Theorem 5.6 is almost equivalent to the proof of Theorem 5.2, except that the problem here is inferring $\theta_h$ separately for each $h \in [H]$.

