# OpenReview forum: "Is O(log N) practical? Near-Equivalence Between Delay Robustness and Bounded Regret in Bandits and RL"
_NeurIPS.cc/2024/Conference — NeurIPS 2024 poster_

### Official Review · Reviewer_3wWt · 2024-06-21

**Soundness:** 3
**Presentation:** 3
**Contribution:** 3
**Rating:** 6
**Confidence:** 1

**Summary:**

The paper explores interactive decision-making scenarios encompassing bandits, contextual bandits, and reinforcement learning, focusing on the concept of regret minimization. It highlights the Graves-Lai constant, where its zero value is crucial for achieving bounded regret in interactive decision-making. This condition, however, may be stringent for practical applications, prompting questions about its feasibility. The study extends this analysis to include robustness against unknown reward delays, termed ϵ-robustness, which measures algorithmic resilience to misspecified delay models. The main finding asserts that achieving ϵ-robustness is impossible for consistent algorithms unless the Graves-Lai constant is zero. The paper contrasts this negative result with positive outcomes in linear reward models, demonstrating that a zero Graves-Lai constant is sufficient for achieving bounded regret without knowledge of delay models. This dual perspective underscores the theoretical and practical implications of the Graves-Lai constant in designing robust learning algorithms for interactive decision-making under uncertainty.

**Strengths:**

The paper makes significant contributions by establishing theoretical results that link the Graves-Lai constant to delay robustness in interactive decision-making scenarios. It introduces the concept of ϵ-delay robustness, which quantifies how learning algorithms perform under ϵ-contaminated delay models.

It provides rigorous theoretical foundations, leveraging concepts from robust statistics and decision theory to analyze the impact of delay model misspecification on learning algorithms.

The problem of attributing delayed rewards to decisions (anonymous feedback) is clearly articulated, which is essential for understanding the challenges addressed by the proposed algorithms.

**Weaknesses:**

While the paper provides theoretical analysis and proofs, empirical validation through simulations or real-world experiments could strengthen the practical relevance of the results. Lack of empirical validation might limit the confidence in how well the theoretical findings translate into actual performance improvements in real interactive decision-making scenarios.

**Questions:**

Can you provide a more intuitive explanation of how delay robustness affects decision-making in interactive systems?

Are there specific conditions under which the proposed delay-robust algorithms may not perform optimally? What are the limitations of the theoretical framework?

How confident are you that the proposed algorithms will perform well in practical scenarios, given the complexities and uncertainties inherent in real-world applications?

**Limitations:**

Explicit discussion of the limitations of the proposed approach and avenues for future research could enhance the paper. Addressing these aspects would provide a more balanced view of the scope and potential of the findings.

---

> ### Author Response · Authors · 2024-08-02
>
> References to the official Rebuttal below:
>
> \[1\]: Foster, Dylan J., et al. "The statistical complexity of interactive decision making." arXiv preprint arXiv:2112.13487 (2021).
>
> \[2\]: Dong, K., & Ma, T. (2023). Asymptotic instance-optimal algorithms for interactive decision making. The Eleventh International Conference on Learning Representations (ICLR)
>
> \[3\]: Wagenmaker, A. J., & Foster, D. J. (2023). Instance-optimality in interactive decision making: Toward a non-asymptotic theory. In The Thirty Sixth Annual Conference on Learning Theory (pp. 1322-1472). PMLR.
>
> \[4\]: Kang, H., & Kumar, P. R. (2023). Recommender system as an exploration coordinator: a bounded O (1) regret algorithm for large platforms. arXiv e-prints, arXiv-2301.
>
> \[5\]: Hao, B., Lattimore, T., & Szepesvari, C. Adaptive exploration in linear contextual bandit. In International Conference on Artificial Intelligence and Statistics (pp. 3536-3545). PMLR. (2020)

---

> ### Author Rebuttal · Authors · 2024-08-02
>
> ### **Answer to comments in "Weaknesses"**
>
> **Comment 1.**
> > While the paper provides theoretical analysis and proofs,
> empirical validation through simulations or real-world experiments
> could strengthen the practical relevance of the results.
> Lack of empirical validation might limit the confidence in
>  how well the theoretical findings translate into actual performance
>  improvements in real interactive decision-making scenarios.
>
> **Author response to Comment 1**:\
> Thank you for your comment. It is true that empirical validation or simulation studies are
> out of our paper's scope as in other papers on DMSO. \
> _(DMSO is a framework that generalizes many different sequential-decision making problems,it is quite uncommon for papers on DMSO to include simulation experiments for particular environments. Examples include all the key papers of this paper:_
> * _The first paper that suggested the concept of DMSO ([Foster, Kakade, Qian, Rakhalin, 2021][1] \[1\])_
> * _The paper that characterizes Graves-Lai coefficient for DMSO ([Dong and Ma, 2023 ][2] \[2\])_
> * _The paper that proposes an instance-optimal algorithm for DMSO ([Wagenmaker and Foster, 2023 ][3] \[3\])_\
> * And so forth, including all other papers on DMSO.
>
> )
>
> However, to help readers to understand the real-world applicability of this paper's theoretical result,
> the new draft will have a paragraph on how the Graves-Lai constant being 0 works in practice.
> *  As discussed in our draft, for linear contextual bandits, the necessary & sufficient condition for bounded regret ([Hao, Lattimore, and Szepesvari 2020][4] \[4\]) can be easily satisfied by having rich enough context space ([Kang and Kumar, 2023][4] \[4\]).
> In [Kang and Kumar, 2023][4] \[4\]'s Spotify context, million daily users can be considered a rich enough context for exploring 60,000 new songs uploaded daily.  That is, we can apply our algorithm to this Spotify music recommendation and exploration example.
>
> ### **Answer to comments in "Questions"**
>
> **Comment 1.**
> > Can you provide a more intuitive explanation of how delay robustness affects decision-making in interactive systems?
>
> **Author response to Comment 1**:\
> To achieve O(log n) regret, you must pull the best arm much more than you pull the other arms. This hinders identification of non-optimal arms, as
> a very small contamination in your knowledge about optimal arm's delay distribution will hinder you from getting a good enough statistical information to help conclude whether a reward is from a non-optimal arm.
>
> **Comment 2.**
> > Are there specific conditions under which the proposed delay-robust algorithms may not perform optimally? What are the limitations of the theoretical framework?
>
> **Author response to Comment 2**:
> * Recall that algorithm proposed is a proof-of-concept algorithm of which purpose is to prove the equivalence of bounded regret and any-level delay robustness.
> To this end, the key assumption we make for this algorithm is that the Graves-Lai constant is 0 (= the iff condition for bounded regret ([Hao, Lattimore and Szepesvari 2021][5] \[5\])).
> We prove $poly(n)$ lower bound for the case when this assumption does not hold. But we don't prove any upper bound for the case when the "Graves-Lai constant being 0" does not hold.
>
> * For non-linear setting, whether our positive result works is an open question.
>
> **Comment 3.**
> > How confident are you that the proposed algorithms will perform well in practical scenarios, given the complexities and uncertainties inherent in real-world applications?
>
> **Author response to Comment 3**:
> * As described above, in real-world online platforms with diverse users, the condition that Graves-Lai constant being 0 easily holds.
> To see how well bounded regret algorithm like ours works in practical recommender systems, see [Kang and Kumar, 2023][4] \[4\].
> * Graves-Lai constant being 0 is not likely to hold in smaller systems.
>
>
> ### **Answer to comments in "Limitations"**
> **Comment 1.**
> >Explicit discussion of the limitations of the proposed approach and avenues for future research could enhance the paper. Addressing these aspects would provide a more balanced view of the scope and potential of the findings.
>
> **Author response to Comment 1**:\
> Thank you for this comment. In discussions section, we will include the following ideas:
> * While this paper proves $poly(n)$ lower bound for the case when "Graves-Lai constant being 0" does not hold,
> upper bound for this case is an open question.
> * The algorithm proposed is a proof-of-concept algorithm of which purpose is to prove the equivalence of bounded regret and any-level delay robustness.
> To this end, the key assumption we make for this algorithm is that the Graves-Lai constant is 0 (= the iff condition for bounded regret ([Hao, Lattimore and Szepesvari 2021][5] \[5\])). While large systems such as Spotify satisfies this condition (a million daily users for 60,000 new songs exploration), smaller systems won't satisfy this condition.
> * As discussed in Section 4.2.1, cross-informativeness is
> the key to achieving the positive result in our paper. For linear cases, we attribute Section 5 to show that cross-informativeness holds when the Graves-Lai constant is 0.
> Whether we can extend the positive result in our paper for other type of models is an open question. Towards this direction, one may want to show that a model's particular structure allows Graves-Lai constant being 0 to imply cross-informativeness.
>
>
>
> [1]: https://arxiv.org/abs/2112.13487
>
> [2]: https://openreview.net/forum?id=oGVu9spZaJJ
>
> [3]: https://proceedings.mlr.press/v195/wagenmaker23a.html
>
> [4]: https://arxiv.org/abs/2301.12571
>
> [5]: https://proceedings.mlr.press/v108/hao20b.html

---

> > ### Comment · Reviewer_3wWt · 2024-08-13
> >
> > I have read the authors' rebuttal and the comments from the other reviewers. I would like to keep my score unchanged.

---

### Official Review · Reviewer_f9VR · 2024-06-26

**Soundness:** 3
**Presentation:** 3
**Contribution:** 2
**Rating:** 5
**Confidence:** 3

**Summary:**

This paper studies anonymous delay (i.e. when it is unknown which trial the delayed reward came from) in interactive decision making. It gives a strongly negative result that if the reward delay distribution is not exactly known and the “Grave-Lai constant” is non-zero then no algorithm has sub-polynomial regret. It also gives a positive result that, for linear-rewards models, the Grave-Lai constant being zero is sufficient for achieving bounded regret with no knowledge of the delay distribution.

**Strengths:**

I think that the negative result (Theorem 4.1) is quite neat although I am not sure of its importance (the other reviewers’ opinions on this will certainly influence my final score).

**Weaknesses:**

Whilst Theorem 4.1 rules out (unless the Grave-Lai constant is zero) sub-polynomial regret with unknown reward distributions, it does not rule out algorithms with very small polynomial regret. A lower bound on a polynomial exponent would be a much better result in my opinion.

The positive result seems limited. As the authors say in the abstract - “as the condition of the Graves-Lai constant being zero may be a strong requirement for many applications, the practical usefulness of pursuing bounded regret (or in this case unknown delay functions) has been questioned”.

Line 7 of Algorithm 1 suggests that F must be finite. If this is true then this is a serious limitation.

**Questions:**

Line 131: what is an $\epsilon$-probability removal?

Line 135: $\nu$ lots a lot like $v$ - I recommend a different letter

---

> ### Author Rebuttal · Authors · 2024-08-02
>
> ### **Answer to comments in "Weaknesses"**
>
> **Comment 1**.
> > "Whilst Theorem 4.1 rules out (unless the Grave-Lai constant is zero) sub-polynomial regret with unknown reward distributions, it does not rule out algorithms with very small polynomial regret. A lower bound on a polynomial exponent would be a much better result in my opinion."
>
>  **Author response to comment 1**:
> * When the Graves-Lai constant is not 0, our first main result shows that no algorithm can achieve sub-polynomial regret. In other words, **"we prove _poly(n)_ lower bound"**, which resonates with your point that _"a lower bound on a polynomial exponent will be nice"_. We greatly appreciate your pointing this out, as we have not yet explicitly stated this sentence in the previous draft.
>
> **Comment 2**.
>
> > "The positive result seems limited. As the authors say in the abstract - “as the condition of the Graves-Lai constant being zero may be a strong requirement for many applications, the practical usefulness of pursuing bounded regret (or in this case unknown delay functions) has been questioned"
>
> **Author response to Comment 2**:\
> Cost of satisfying condition of Graves-Lai constant being zero indeed questions the usefulness of bounded regret algorithm design, but **the cost we should pay is not too much impractical**. As discussed in our draft, for linear contextual bandits, for example, the necessary & sufficient condition for bounded regret ([Hao, Lattimore, and Szepesvari 2020][2] \[2\]) can be easily satisfied by having rich enough context space ([Kang and Kumar, 2023][3] \[3\]). For example, in [Kang and Kumar, 2023][3] \[3\]'s Spotify context, million daily users can be considered a rich enough context for exploring 60,000 new songs uploaded daily.
>
> **Comment 3**.
>
> > "Line 7 of Algorithm 1 suggests that F must be finite. If this is true then this is a serious limitation."
>
> **Author response to Comment 3**:\
> Thank you for your suggestion. We modified our Algorithm 1 pseudocode to remove the error.\
> [Link to the new Algorithm 1 pseducode's image][4]
>
>
>
> ### **Answer to comments in "Questions"**
>
> **Comment 1.**
> > "Line 131 : what is an $\epsilon$-probability removal?"
>
> **Author respoonse to Comment 1**:\
> $\epsilon$-probability removal from a distribution is also called "Subtractive Contamination" in robust statistics literature. The definition is as follows:\
> _Definition ($\epsilon$-probability removal or Subtractive Contamination)_\
> _Given a parameter $0<\epsilon<1$ and a distribution $D$ on inliers, we say that one can sample from $D$ with $\epsilon$-subtractive contamination if the following holds: for some event $R$ with probability $1-\epsilon$, one can obtain independent samples from the distribution of $D$ conditioned on $R$. In other words, with probability $\epsilon$, the event $R^c$ occurs and these samples are removed from the data stream. This allows an adversary to remove an $\epsilon$-fraction of inlier samples._
>
> **Comment 2.**
>
> > Line 135: $\nu$ lots a lot like $v$ - I recommend a different letter.
>
> **Author response to Comment 2**:\
> The new draft now uses $\mu$.\
> (Please note that we cannot show this change, as NeurIPS does not allow the pdf draft replacement in openreview)
>
> [1]: https://epubs.siam.org/doi/abs/10.1137/S0363012994275440
>
> [2]: https://proceedings.mlr.press/v108/hao20b.html
>
> [3]: https://arxiv.org/abs/2301.12571
>
> [4]: https://postimg.cc/VSdHXhf7
>
>
> \[1\]: Graves, T. L., & Lai, T. L. (1997). Asymptotically efficient adaptive choice of control laws incontrolled markov chains. SIAM journal on control and optimization, 35(3), 715-743.
>
> \[2\]: Hao, B., Lattimore, T., & Szepesvari, C. (2020). Adaptive exploration in linear contextual bandit. In International Conference on Artificial Intelligence and Statistics (pp. 3536-3545). PMLR.
>
> \[3\]: Kang, H., & Kumar, P. R. (2023). Recommender system as an exploration coordinator: a bounded O (1) regret algorithm for large platforms. arXiv e-prints, arXiv-2301.

---

> > ### Comment · Reviewer_f9VR · 2024-08-12
> >
> > I do not think that your response to comment 1 is correct. For instance, we could have, for every $\epsilon>0$, an algorithm with a regret of $\mathcal{O}(T^\epsilon)$. This means there is no polynomial lower bound. This does not mean that there exists an algorithm with regret $o(T^\epsilon)$ for every $\epsilon>0$. Hence, proving that there exists no algorithm with sub-polynomial regret is not the same thing as proving that there exists a polynomial lower bound on the regret. In any case, my question is asking if you can give the lower bound on the exponent (if one exists), which I guess would be problem dependent.
> >
> > I don't understand your response to comment 2 - could you please rephrase?
> >
> > In regards to comment 3 - in your new algorithm (for infinite F) how computationally hard is it to find such a g?

---

> ### Author Response · Authors · 2024-08-12
>
> **Comment 1.**
> > I do not think that your response to comment 1 is correct. For instance, we could have, for every $\epsilon>0$, an algorithm with a regret of $\mathcal{O}\left(T^\epsilon\right)$. This means there is no polynomial lower bound. This does not mean that there exists an algorithm with regret $o\left(T^\epsilon\right)$ for every $\epsilon>0$. Hence, proving that there exists no algorithm with sub-polynomial regret is not the same thing as proving that there exists a polynomial lower bound on the regret. In any case, my question is asking if you can give the lower bound on the exponent (if one exists), which I guess would be problem dependent.
>
> **Answer to comment 1**.\
> I see your point. Although this paper's scope does not include an answer to what you are asking for, trying to answer that will be a really nice future research direction. **We will state this point in the discussion section, as it demonstrates a sharp boundary of our contribution**. This paper's focus is indeed on rejecting the popular notion of "consistent" algorithm design, which assure _uniform_ $o(n^p)$ regret for all $p>0$ and for all problem instances. As you pointed out, if we just say that there is no algorithm that satisfies $n^p$ regret for some $p$ we are wrong; we can at most say that there is no algorithm that satisfy $n^p$ regret for all $p$ for all problem instances. (i.e. it is kind of _uniform_ $poly(T)$ lower bound, **which is much weaker than $poly(T)$ lower bound**). Your point is clearly legitimate, as our result only negates the notion of consistency. Thank you very much for your comment.
>
> **Comment 2.**
> >I don't understand your response to comment 2 - could you please rephrase?
>
> **Answer to Comment 2.**\
> In linear contextual bandits, for example, Graves-Lai constant being 0 can be easily satisfied in large enough systems; it only requires some constant times $klogk$ daily Spotify users (=# of contexts) to satisfy the constraint of graves-Lai constant being 0, if the number of new songs (=# of arms) uploaded daily is $k$ ([Kang and Kumar 2023][1]). As daily new songs to explore are around 60,000 and $60,000 \times \log 60,000 \le 300,000$ daily users, we can apply any algorithm that requires Graves-Lai constant being 0 for Spotify (which has more than $100m$ daily users).
>
> [1]: https://arxiv.org/abs/2301.12571
>
> **Comment 3.**
> >In regards to comment 3 - in your new algorithm (for infinite F) how computationally hard is it to find such a g?
>
> **Answer to Comment 3.**\
> We really appreciate your pointing this out. Although $\mathcal{F}_n$ in our new algorithm can be easily computed in some problems like bandits, **we may definitely need to assume an existence of an oracle who can find an instance $g\in \mathcal{F}_n$ in general to abstract the computational complexity**. Like [Wagenmaker and Foster 2023][2] does, we will add an assumption on this and state that "We emphasize that the focus of this work is primarily statistical, and leave addressing the computational challenges for specific problems for future work." Thank you for improving this work.
>
> [2]: https://proceedings.mlr.press/v195/wagenmaker23a.html

---

### Official Review · Reviewer_uxdM · 2024-06-27

**Soundness:** 3
**Presentation:** 2
**Contribution:** 3
**Rating:** 5
**Confidence:** 4

**Summary:**

This paper studies the relationship between bounded regret and delay robustness in interactive decision-making, which captures bandits, contextual bandits, and reinforcement learning. The authors show that the Graves-Lai constant being zero is necessary for achieving delay model robustness when reward delays are unknown. On the other hand, it is also shown that the Graves-Lai constant being zero is sufficient for achieving bounded regret without delay model knowledge for linear reward models.

**Strengths:**

1. The paper introduces a novel connection between bounded regret and delay robustness, offering a fresh perspective on interactive decision-making.
2. The paper presents results in both directions, although positive results are limited to linear models.

**Weaknesses:**

Indeed, the requirement for the Graves-Lai constant to be exactly zero is exceedingly strong. In the context of linear bandits, it necessitates that the set of optimal actions spans the entire action space, a condition that is nearly unattainable in practical applications. This consideration could potentially diminish the paper's overall significance.

**Questions:**

1. The positive results are specific to linear reward models. Could you please elaborate on the challenges in extending these results to other types of models?

2. What if the reward delay distribution is precisely known?

3. Has there been any prior research in bandit or reinforcement learning that investigates the connection between regret and delay?

4. Minor typo on Line 253: "indicates" -> "indicate".

**Limitations:**

This theoretical paper may have limited direct societal impact.

---

> ### Author Response · Authors · 2024-08-01
>
> References to the official Rebuttal below:
>
> \[1\]: Hao, B., Lattimore, T., & Szepesvari, C. Adaptive exploration in linear contextual bandit. In International Conference on Artificial Intelligence and Statistics (pp. 3536-3545). PMLR. (2020)
>
> \[2\]:  Kang, H., & Kumar, P. R. (2023). Recommender system as an exploration coordinator: a bounded O (1) regret algorithm for large platforms. arXiv e-prints, arXiv-2301.
>
> \[3\]: Pike-Burke, Ciara, et al. "Bandits with delayed, aggregated anonymous feedback." International Conference on Machine Learning. PMLR, 2018.
>
> \[4\]: Thune, Tobias Sommer, Nicolò Cesa-Bianchi, and Yevgeny Seldin. "Nonstochastic multiarmed bandits with unrestricted delays." Advances in Neural Information Processing Systems 32 (2019).
>
> \[5\]: Wu, Han, and Stefan Wager. "Thompson sampling with unrestricted delays." Proceedings of the 23rd ACM Conference on Economics and Computation. 2022.
>
> \[6\]: Jin, Tiancheng, et al. "Near-optimal regret for adversarial mdp with delayed bandit feedback." Advances in Neural Information Processing Systems 35 (2022): 33469-33481.
>
> \[7\]: Masoudian, S., Zimmert, J., & Seldin, Y. A Best-of-both-worlds Algorithm for Bandits with Delayed Feedback with Robustness to Excessive Delays. In ICML 2024 Workshop: Foundations of Reinforcement Learning and Control
>
> \[8\]: Zimmert, J., & Seldin, Y. (2020, June). An optimal algorithm for adversarial bandits with arbitrary delays. In International Conference on Artificial Intelligence and Statistics (pp. 3285-3294). PMLR.

---

> ### Author Rebuttal · Authors · 2024-08-02
>
> ### **Answer to comments in "Weaknesses"**
>
> **Comment 1.**
> > Indeed, the requirement for the Graves-Lai constant to be exactly zero is exceedingly strong.
> In the context of linear bandits, it necessitates that the set of optimal actions spans the entire action space, a condition that is nearly unattainable in practical applications.
> This consideration could potentially diminish the paper's overall significance.
>
> **Author response to Comment 1**:
> * **Because the requirement for bounded regret is indeed exceedingly strong, our negative result shines.**: We show that the consistent algorithm design regime, one of the most popular algorithm design regime, may be impractical under anonymous reward delays, as Graves-Lai constant to be exactly zero is exceedingly strong. **In other words, we prove reduction of consistent algorithms with delayed rewards to algorithms with bounded regret in DMSO**.
> * Cost of satisfying condition of Graves-Lai constant being zero indeed questions the usefulness of bounded regret algorithm design, but **the cost we should pay is not too much impractical in practice**. As discussed in our draft, for linear contextual bandits, for example, the necessary & sufficient condition for bounded regret ([Hao, Lattimore, and Szepesvari 2020][1] \[1\]) can be easily satisfied by having rich enough context space ([Kang and Kumar, 2023][2] \[2\]). For example, in [Kang and Kumar, 2023][2] \[2\]'s Spotify context, million daily users can be considered a rich enough context for exploring 60,000 new songs uploaded daily. That is, we can apply our algorithm to this Spotify music recommendation and exploration example.
>
>
> ### **Answer to comments in "Questions"**
>
> **Comment 1.**
> > The positive results are specific to linear reward models.
> Could you please elaborate on the challenges in extending these results to other types of models?
>
> **Author response to Comment 1**:\
> Thank you for this question. As discussed in Section 4.2.1, cross-informativeness is
> the key to achieving such positive result. For linear cases, we attribute Section 5 to show that cross-informativeness holds when the Graves-Lai constant is 0.
> To extend the positive result in our paper for other type of models, one can show that the model's particular structure allows Graves-Lai constant being 0 to imply cross-informativeness.
> We will add this answer to the discussions section.
>
> **Comment 2.**
> > What if the reward delay distribution is precisely known?
>
> **Author response to Comment 2**:
> * As discussed in 1.1. Related work, we can then apply [Pike-Burke et al., 2018][3] \[3\], which requires an assumption that the mean of delay distribution is precisely known (which cannot be achieved under $\epsilon$-contamination however small $\epsilon$ is).
> * Of course, knowing _anonymous_ reward's delay distribution precisely is highly unrealistic.
>
> **Comment 3.**
> > Has there been any prior research in bandit or reinforcement learning that investigates the connection between regret and delay?
>
> **Author response to Comment 3**:\
> Thank you for your question. We believe that we were not clear enough about the fact that this is the first
> paper that approaches _anonymous_ delay in rewards through the lens of delay robustness.
>
> For the papers that discuss robustness/unrestricted delay for _non-anonymous_ delays (i.e., the setup where agents can associate each delayed reward to the arm it is from),
> we are adding a new separate paragraph with 10-15 papers in the related work section, including:
>
> * [Thune, Cesa-Bianchi, and Seldin (2019)][4] \[4\]
> * [Wu, Ha, and Wager (2022)][5] \[5\]
> * [Jin, Lancewicki, Luo, Mansour, Rosenberg (2022)][6] \[6\]
> * [Masoudian, Zimmert, and Seldin (2024)][7] \[7\]
> * [Zimmert and Seldin (2020)][8] \[8\]
> * ...
>
> **Comment 4.**
> > Minor typo on Line 253: "indicates" -> "indicate".
>
> **Author response to Comment 4**:\
> Thank you for pointing this out. We have fixed the typo.
>
>
> [1]: https://proceedings.mlr.press/v108/hao20b.html
> [2]: https://arxiv.org/abs/2301.12571
> [3]: https://arxiv.org/abs/1709.06853
> [4]: https://proceedings.neurips.cc/paper/2019/hash/0e4f5cc9f4f3f7f1651a6b9f9214e5b1-Abstract.html
> [5]: https://dl.acm.org/doi/abs/10.1145/3490486.3538376
> [6]: https://proceedings.neurips.cc/paper_files/paper/2022/hash/d850b7e0cdc7f1c0820c6ad85405ae94-Abstract-Conference.html
> [7]: https://openreview.net/forum?id=aLgJssbizV
> [8]: https://proceedings.mlr.press/v108/zimmert20a.html

---

> > ### Comment · Reviewer_uxdM · 2024-08-13
> >
> > Thank you for your rebuttal. After considering the rebuttal, I have decided to keep my original rating.

---

### Official Review · Reviewer_ykEi · 2024-07-04

**Soundness:** 3
**Presentation:** 2
**Contribution:** 2
**Rating:** 5
**Confidence:** 3

**Summary:**

The paper investigates interactive decision-making in bandits, contextual bandits, and reinforcement learning, focusing on the Graves-Lai constant's role in achieving bounded regret. It establishes that a zero Graves-Lai constant is necessary and sufficient for bounded regret, but questions its practical utility due to its stringent requirements. The study shows that $\epsilon$-robustness in delay model robustness cannot be achieved if the Graves-Lai constant is non-zero, presenting a negative result for consistent algorithms. However, it offers a positive result for linear rewards models, indicating that a zero Graves-Lai constant is sufficient for bounded regret without delay model knowledge, balancing efficiency and robustness.

**Strengths:**

The paper investigated some interesting questions.

**Weaknesses:**

1. The related work on robustness seems not comprehensive.
2. There is no experiment.
3. The authors didn't give proper citations in definitions or assumptions and so on.

**Questions:**

1. Is the definition of "the sub-optimality gap of decision" also used in the previous paper? If yes, give the citation.
2. Is the assumption of "Realizability" generally used in other work? If yes, give the citation.
3. The lower bound is studied by other work. What is the explicit form of the lower bound and can you derive the instance-independent lower bound?
4. What is the application of your algorithm and can you run some experiments?

**Limitations:**

There is no experiment.

---

> ### Author Response · Authors · 2024-08-01
>
> References to the official Rebuttal below:
>
> \[1\]: Foster, Dylan J., et al. "The statistical complexity of interactive decision making." arXiv preprint arXiv:2112.13487 (2021).
>
> \[2\]: Dong, K., & Ma, T. (2023). Asymptotic instance-optimal algorithms for interactive decision making. The Eleventh International Conference on Learning Representations (ICLR)
>
> \[3\]: Wagenmaker, A. J., & Foster, D. J. (2023). Instance-optimality in interactive decision making: Toward a non-asymptotic theory. In The Thirty Sixth Annual Conference on Learning Theory (pp. 1322-1472). PMLR.
>
> \[4\]: Chen, F., & Mei S., & Chen F. (2024). Unified Algorithms for RL with Decision-Estimation Coefficients: PAC, Reward-Free, Preference-Based Learning, and Beyond
>
> \[5\]: Foster, D. J., Golowich, N., & Han, Y. (2023, July). Tight guarantees for interactive decision making with the decision-estimation coefficient. In The Thirty Sixth Annual Conference on Learning Theory (pp. 3969-4043). PMLR.
>
> \[6\]: Foster, D. J., Foster, D. J., Golowich, N., Qian, J., Rakhlin, A., & Sekhari, A. (2023). Model-free reinforcement learning with the decision-estimation coefficient. Advances in Neural Information Processing Systems, 36.
>
> \[7\]: Foster, D. J., Han, Y., Qian, J., & Rakhlin, A. (2024). Online estimation via offline estimation: An information-theoretic framework. arXiv preprint arXiv:2404.10122.
>
> \[8\]: Thune, Tobias Sommer, Nicolò Cesa-Bianchi, and Yevgeny Seldin. "Nonstochastic multiarmed bandits with unrestricted delays." Advances in Neural Information Processing Systems 32 (2019).
>
> \[9\]: Wu, Han, and Stefan Wager. "Thompson sampling with unrestricted delays." Proceedings of the 23rd ACM Conference on Economics and Computation. 2022.
>
> \[10\]: Jin, Tiancheng, et al. "Near-optimal regret for adversarial mdp with delayed bandit feedback." Advances in Neural Information Processing Systems 35 (2022): 33469-33481.
>
> \[11\]: Masoudian, S., Zimmert, J., & Seldin, Y. A Best-of-both-worlds Algorithm for Bandits with Delayed Feedback with Robustness to Excessive Delays. In ICML 2024 Workshop: Foundations of Reinforcement Learning and Control
>
> \[12\]: Zimmert, J., & Seldin, Y. (2020, June). An optimal algorithm for adversarial bandits with arbitrary delays. In International Conference on Artificial Intelligence and Statistics (pp. 3285-3294). PMLR.
>
> \[13\]: Agarwal, Alekh, et al. "Contextual bandit learning with predictable rewards." Artificial Intelligence and Statistics. PMLR, (2012).
>
> \[14\]: Du, Simon, et al. "Bilinear classes: A structural framework for provable generalization in rl." International Conference on Machine Learning. PMLR, (2021).
>
> \[15\]:  Kang, H., & Kumar, P. R. (2023). Recommender system as an exploration coordinator: a bounded O (1) regret algorithm for large platforms. arXiv e-prints, arXiv-2301.
>
> \[16\]: Hao, B., Lattimore, T., & Szepesvari, C. Adaptive exploration in linear contextual bandit. In International Conference on Artificial Intelligence and Statistics (pp. 3536-3545). PMLR. (2020)

---

> ### Author Rebuttal · Authors · 2024-08-02
>
> ### **Answer to comments in "Weaknesses"**
>
> **Comment 1**.
>
> > "There is no experiment."
>
> **Author response to comment 1**:\
> AS DMSO is a framework that generalizes many different sequential-decision making problems
>  such as bandits, contextual bandits, and reinforcement learning and so forth, it is quite uncommon for papers on DMSO to include
> simulation experiments for particular environments. Examples include all the key papers of this paper:
> * The first paper that suggested the concept
> of DMSO ([Foster, Kakade, Qian, Rakhalin, 2021][1] \[1\])
> * The paper that characterizes Graves-Lai coefficient for DMSO ([Dong and Ma, 2023 ][2] \[2\])
> * The paper that proposes an instance-optimal algorithm for DMSO ([Wagenmaker and Foster, 2023 ][3] \[3\])
>
> and all other related papers on DMSO:
> * [Chen, Mei and Bai (2024)][4] \[4\]
> * [Foster, Golowich and Han (2023)][5] \[5\]
> * [Foster, Golowich, Qian and Rakhlin (2023)][6] \[6\]
> * [Foster, Han, Qian and Rakhlin (2024)][7] \[7\]
>
> **Comment 2**.
>
> > "The related work on robustness seems not comprehensive."
>
> **Author response to Comment 2**:\
> Thank you for pointing this out. We believe that we were not clear enough about the fact that this is the first
> paper that approaches _anonymous_ delay in rewards through the lens of robustness.
>
> For the papers that discuss robustness/unrestricted delay for _non-anonymous_ delays (i.e., the setup where agents can associate each delayed reward to the arm it is from),
> we are adding a new separate paragraph with 10-15 papers in the related work section, including:
>
> * [Thune, Cesa-Bianchi, and Seldin (2019)][8] \[8\]
> * [Wu, Ha, and Wager (2022)][9] \[9\]
> * [Jin, Lancewicki, Luo, Mansour, Rosenberg (2022)][10] \[10\]
> * [Masoudian, Zimmert, and Seldin (2024)][11] \[11\]
> * [Zimmert and Seldin (2020)][12] \[12\]
> * ...
>
> **Comment 3.**
> > The authors didn't give proper citations in definitions or assumptions and so on.
>
> **Author response to Comment 3**:\
> Thank you for pointing this out and leaving detailed instruction in **Questions** section as questions. We will answer your questions in the next section.
>
>
> ### **Answer to comments in "Questions"**
> **Comment 1.**
> >Is the definition of "the sub-optimality gap of decision" also used in the previous paper? If yes, give the citation.
>
> **Author response to Comment 1**:\
> This is a kind of folklore definition used in almost every bandit/reinforcement learning
> papers that deals with the concept of regret. Specifically for a decision's sub-optimality gap,
> we can cite [Foster, Kakade, Qian, Rakhalin, 2021][1] \[1\], as this paper defines "decision" in a DMSO framework and also the sub-optimality gap of a decision accordingly.
>
> **Comment 2.**
> > Is the assumption of "Realizability" generally used in other work? If yes, give the citation.
>
> **Author response to Comment 2**:\
> Thank you for pointing this out; we will cite the following three papers.
> * [Agarwal, Dudik, Kale, Langford and Schapire, 2012][13] \[13\]
> * [Du, Kakade, Lee, Lovett, Mahajan, Sun and Wang, 2021][14] \[14\]
> * [Foster, Kakade, Qian, Rakhalin, 2021][1] \[1\])
>
> **Comment 3.**
> > The lower bound is studied by other work. What is the explicit form of the lower bound and can you derive the instance-independent lower bound?
>
> **Author response to Comment 3**:
> * When the Graves-Lai constant is 0, the lower bound is clearly O(1), as we proved that there is an algorithm
> that achieves the upper bound of O(1).
> * When the Graves-Lai constant is not 0, our first main result shows that no algorithm can achieve sub-polynomial regret. In other words, "we prove _poly(n)_ lower bound". We greatly appreciate your pointing out this, as we have not yet explicitly stated this sentence in the previous draft.
>
> **Comment 4.**
> > What is the application of your algorithm and can you run some experiments?
>
> **Author response to Comment 4**:\
> While the algorithm proposed is a proof-of-concept algorithm of which purpose is to
> prove the equivalence of bounded regret and any-level delay robustness, you can apply this algorithm for real-world platforms.
> As discussed in our draft, for linear contextual bandits, for example, the necessary & sufficient condition for bounded regret ([Hao, Lattimore, and Szepesvari 2020][16] \[16\]) can be easily satisfied by having rich enough context space ([Kang and Kumar, 2023][15] \[15\]).
> In [Kang and Kumar, 2023][15] \[15\]'s Spotify context, million daily users can be considered a rich enough context for exploring 60,000 new songs uploaded daily. That is, we can apply our algorithm to this Spotify music recommendation and exploration example.
>
> For the author response related to experiments, please see above.
>
>
> [1]: https://arxiv.org/abs/2112.13487
>
> [2]: https://openreview.net/forum?id=oGVu9spZaJJ
>
> [3]: https://proceedings.mlr.press/v195/wagenmaker23a.html
>
> [4]: https://arxiv.org/abs/2209.11745
>
> [5]: https://proceedings.mlr.press/v195/foster23b.html
>
> [6]: https://proceedings.neurips.cc/paper_files/paper/2023/hash/3fcd0f8747f9217c6dbc45ed138b1fde-Abstract-Conference.html
>
> [7]: https://arxiv.org/abs/2404.10122
>
> [8]: https://proceedings.neurips.cc/paper/2019/hash/0e4f5cc9f4f3f7f1651a6b9f9214e5b1-Abstract.html
>
> [9]: https://dl.acm.org/doi/abs/10.1145/3490486.3538376
>
> [10]: https://proceedings.neurips.cc/paper_files/paper/2022/hash/d850b7e0cdc7f1c0820c6ad85405ae94-Abstract-Conference.html
>
> [11]: https://openreview.net/forum?id=aLgJssbizV
>
> [12]: https://proceedings.mlr.press/v108/zimmert20a.html
>
> [13]: https://proceedings.mlr.press/v22/agarwal12
>
> [14]: https://proceedings.mlr.press/v139/du21a.html
>
> [15]: https://arxiv.org/abs/2301.12571
>
> [16]: https://proceedings.mlr.press/v108/hao20b.html

---

> > ### Comment · Reviewer_ykEi · 2024-08-08
> >
> > Thank you for the reply. The responses are satisfactory. Please add these works in your next version. I am raising my score by +1.

---

### Official Review · Reviewer_bj5z · 2024-07-16

**Soundness:** 3
**Presentation:** 2
**Contribution:** 3
**Rating:** 6
**Confidence:** 3

**Summary:**

The paper considers the problem of regret minimization under delayed rewards. The paper considers the setting where the Decision-making with Structured Observations  (DMSO) setting and asks when the learner can achieve logarithmic regret when the reward signal is delayed and we only have an estimate of the delay distribution up to precision \epsion within the given model class.

The main results are twofold:
1. They show that logarithmic regret under \eps-misspecified delay distribution is possible (no matter what is \eps) only if a certain algebraic quantity called the graves-lai constant is 0 for all models in the given family. Interestingly, the same condition is necessary for obtaining constant regret bounds in DMSO.
2. They provide an upper bound showing that for learning settings for which the the graves-lai constant is 0, and an additional condition called cross-informativeness holds, then one can obtain an upper bound on the regret under delayed reward feedback (with \eps-contaminated delay distribution).

Overall I like the paper as it draws interesting connections between two seemingly different learning settings.

**Strengths:**

- Interesting connection between two seemingly unrelated topics
- Applications to linear contextual bandits and linear MDPs given

**Weaknesses:**

Weakness:

I think the overall exposition can be improved a bit. In particular, it was not clear to me for a long time whether the agent also received the time step for which the reward corresponds to, when it received the delayed reward at a later time step. Other minor issues are:
- Theorem 4.11 should provide a regret bound, if possible.
- Definition 5.4 is also called the Linear Bellman Complete setting. It might be worth adding this phrase for completeness.

Minor typo:
- "for of" at the end of page 5 last paragraph
- First paragraph of page 5 says that d_{TV}(D, D^) >= \eps implies that |E[D] - E[D^]| > 0 (should be >=0 I think).
- Section 4.2 title -- remove "when"
- Section 4.2.2 first line has redundancy

**Questions:**

1. Given the close connection, is there a direction reduction of consistent algorithms with delayed rewards to algorithms with bounded regret in DMSO?
2. The definition of consistency in Defn 2.2 seems to only consider algorithms that have at most logarithmic regret. Can we handle \eps-contamination in reward delay distribution, in the absence of the graves-lai constant being 0, if we are OK with some sort of sublinear regret? It seems to me right now that this notion of consistency is too strong.

---

> ### Author Rebuttal · Authors · 2024-08-02
>
> ### **Answer to comments in "Weaknesses"**
>
> **Comment 1.**
> > It was not clear to me for a long time whether the agent also received the time step for which the reward corresponds to, when it received the delayed reward at a later time step.
>
> **Author response to Comment 1**:\
> Thank you for pointing this out. We believe two changes we made will address this:
> * Instead of "unknown reward delays", we now use "anonymous delayed rewards" to avoid ambiguity and improve representation.
> * We added a clarifying sentence in the introduction that _"the agent never observes the period information for which each reward corresponds to, even after it receives the delayed reward at the later time step."_ Thank you for your suggestion.
>
> **Comment 2.**
> > Theorem 4.11 should provide a regret bound, if possible.
>
> **Author response to Comment 2**:
>
> _(Modified Theorem 4.11)_:
> *  _Under Assumption 4.7 and 4.8, the algorithm ST2C, which does not require any knowledge of the delay distribution model, achieves bounded regret. More precisely, the regret is bounded by $\Delta(1+5\frac{4 c^4 e^{-2} }{\mathcal{W}\left(2 c^2\right)^2}) \frac{\pi^2}{6}$, where $\mathcal{W}$ denote the principal branch lambert W function (which is an increasing function)._
>
> , where $\Delta$ is the maximum per-period mean reward difference among decisions and $c$ is from Assumption 4.8:
>
> _(Modififed Assumption 4.8)_:
> * _For all $f, g \in \mathcal{F}, \pi \in \Pi$, and $o \in \mathcal{O}, |\log \frac{f\pi}{g\pi} |<c$ for some $c>0$._
>
> **Comment 3.**
> > Definition 5.4 is also called the Linear Bellman Complete setting. It might be worth adding this phrase for completeness.
>
> **Author response to Comment 3**:\
> Thank you for your suggestion. We updated this part accordingly.
>
> **Comment 4.**
> > Fix minor typo.
>
> **Author response to Comment 4**:\
> Thank you for pointing them out. We fixed them accordingly.
>
> ### **Answer to comments in "Questions"**
> **Comment 1.**
> > Given the close connection, is there a direct reduction of consistent algorithms with delayed rewards to algorithms with bounded regret in DMSO?
>
> **Author response to Comment 1**:\
> The phrase _"reduction of consistent algorithms with delayed rewards to algorithms with bounded regret in DMSO"_ correctly depicts our contribution. We will add it to the conclusion section.
> In other words, we prove $poly(n)$ lower bound for the case when we cannot achieve bounded regret.
>
> **Comment 2.**
> >The definition of consistency in Defn 2.2 seems to only consider algorithms that have at most logarithmic regret. Can we handle $\epsilon$-contamination in reward delay distribution, in the absence of the graves-lai constant being 0, if we are OK with some sort of sublinear regret? It seems to me right now that this notion of consistency is too strong.
>
> **Author response to Comment 2**:\
> You are absolutely right. While this paper proves $poly(n)$ lower bound for the case when "Graves-Lai constant being 0" does not hold,
> upper bound for this case is an open question.

---

### Author Rebuttal · Authors · 2024-08-03

## 1. Overall comments and thank you response
We first want to thank all reviewers for putting enormous efforts in reviewing this paper. \
We are happy to hear that there was no major issue found by the reviewers, while we were told by all 5 reviewers that this paper
* makes significant contributions by establishing theoretical results and provides rigorous theoretical foundations leveraging concepts from robust statistics and decision theory, while the problem is clearly articulated, which is essential for understanding the challenges addressed by the proposed algorithms. _(Reviewer 3wWt)_
* introduces a novel connection between bounded regret and delay robustness _(Reviewer uxdM)_
* draws interesting connections between two seemingly different learning settings _(Reviewer bj5z)_
* investigated some interesting questions _(Reviewer ykEi)_
* presents a negative result that is quite neat _(Reviewer f9VR)_

### **For each of the reviews, we have submitted a separate author rebuttal below.**

## 2. Summary of the author responses on weaknesses and questions by multiple reviewers
**Point 1.**
> "incomprehensiveness of related works on robustness/unrestricted delay" (by Reviewer ykEi and uxdM)

**Answer to Point 1**: \
We believe that we were not clear enough about the fact that this is the first paper that approaches _anonymous_ delay in rewards through the lens of delay robustness. \
For the papers that discuss robustness/unrestricted delay for _non-anonymous_ delays (i.e., the setup where agents can associate each delayed reward to the arm it is from), we are adding a new separate paragraph with 10-15 papers in the related work section, including:
* [Thune, Cesa-Bianchi, and Seldin (2019)][4]
* [Wu, Ha, and Wager (2022)][5]
* [Jin, Lancewicki, Luo, Mansour, Rosenberg (2022)][6]
* [Masoudian, Zimmert, and Seldin (2024)][7]
* [Zimmert and Seldin (2020)][8]

**Point 2.**
> "There is no experiment" (by Reviewer ykEi and 3wWt)

**Answer to Point 2**: \
AS DMSO is a theoretical framework that generalizes many different sequential-decision making problems
 such as bandits, contextual bandits, and reinforcement learning and so forth, **it is not a common practice for papers on DMSO to include
simulation experiments for particular environments.** Examples include all the key papers of this paper:
* The first paper that suggested the concept
of DMSO ([Foster, Kakade, Qian, Rakhalin, 2021][1])
* The paper that characterizes Graves-Lai coefficient for DMSO ([Dong and Ma, 2023][2])
* The paper that proposes an instance-optimal algorithm for DMSO ([Wagenmaker and Foster, 2023][3])

and all other papers on DMSO:
* [Chen, Mei and Bai (2024)][9]
* [Foster, Golowich and Han (2023)][10]
* [Foster, Golowich, Qian and Rakhlin (2023)][11]
* [Foster, Han, Qian and Rakhlin (2024)][12]

**Point 3.**
> "The requirement for bounded regret is indeed exceedingly strong, as pointed out by authors. Doesn't it potentially diminish the paper's overall significance? Is it practical?" (Reviewer f9VR, uxdM, bj5z)

**Answer to Point 3**:
* **Because the requirement for bounded regret is indeed exceedingly strong, our negative result shines**; we show that _consistent (i.e., $\log n$ regret for all instances) algorithm design (e.g., ([Dong and Ma, 2023][2] and [Wagenmaker and Foster, 2023][3])  may not be practical_ under anonymous delayed rewards, as _we prove reduction of consistent algorithms with anonymous delayed rewards to algorithms with bounded regret_ in DMSO. That is, we prove $poly(n)$ lower bound for the case when bounded regret cannot be achieved.
* The algorithm proposed for positive result is a proof-of-concept algorithm of which purpose is to prove the equivalence of bounded regret and any-level delay robustness. To this end, the key assumption we make for this algorithm is that the Graves-Lai constant is 0 (= the iff condition for bounded regret ([Hao, Lattimore and Szepesvari 2021][14])). **It is not hard for systems with large enough user pool such as Spotify to satisfy this strong requirement** (a million daily users are enough for 60,000 new songs exploration ([Kang and Kumar, 2023][13]). Therefore, we can apply our proposed algorithm in large digital platforms such as Spotify. **However, smaller systems won't satisfy this requirement**.

## 3. Attached pdf
The attached pdf includes the following information:
* Updated Algorithm 1 with improved representation, to fix the issue raised by reviewer f9VR

[1]: https://arxiv.org/abs/2112.13487
[2]: https://openreview.net/forum?id=oGVu9spZaJJ
[3]: https://proceedings.mlr.press/v195/wagenmaker23a.html
[4]: https://proceedings.neurips.cc/paper/2019/hash/0e4f5cc9f4f3f7f1651a6b9f9214e5b1-Abstract.html
[5]: https://dl.acm.org/doi/abs/10.1145/3490486.3538376
[6]: https://proceedings.neurips.cc/paper_files/paper/2022/hash/d850b7e0cdc7f1c0820c6ad85405ae94-Abstract-Conference.html
[7]: https://openreview.net/forum?id=aLgJssbizV
[8]: https://proceedings.mlr.press/v108/zimmert20a.html
[9]: https://arxiv.org/abs/2209.11745
[10]: https://proceedings.mlr.press/v195/foster23b.html
[11]: https://proceedings.neurips.cc/paper_files/paper/2023/hash/3fcd0f8747f9217c6dbc45ed138b1fde-Abstract-Conference.html
[12]: https://arxiv.org/abs/2404.10122
[13]: https://arxiv.org/abs/2301.12571
[14]: https://proceedings.mlr.press/v108/hao20b.html

---

### Decision · Program_Chairs · 2024-09-25

**Decision:**

Accept (poster)

**Comment:**

This paper received rather positive reviews, but none of the reviewers was that excited. So I read the paper myself, and I found it interesting and quite original.

After discussion, the only reviewer that was slightly negative changed the grade to 5, hence I am happy to recommend acceptance !